



# Groundwater storage dynamics and climate variability in the Lower Kutai Basin of Indonesia: reconciling GRACE ΔGWS to piezometry

Arifin[1,2], Richard G. Taylor[1], Mohammad Shamsudduha[3], Agus M. Ramdhan[2]

[1]Department of Geography, University College London, London, WC1E 6BT, UK
[2]Department of Geology, Bandung Institute of Technology, Bandung, 40132, Indonesia
[3]Department of Risk and Disaster Reduction, University College London, London, WC1E 6BT, UK

*Correspondence to*: Arifin (arifin.arifin.22@ucl.ac.uk)

**Abstract.** Groundwater is considered a climate-resilient source of freshwater yet its long-term response to climate variability remains poorly understood in environments with limited ground-based monitoring networks. In the Lower Kutai Basin (LKB)
where Indonesia's new capital (Nusantara) is under development, we examine evidence from Gravity Recovery and Climate Experiment (GRACE) satellite data, global-scale models, precipitation records, and in situ piezometric observations to investigate groundwater storage changes (ΔGWS) over the last two decades. GRACE-derived terrestrial water storage anomalies (ΔTWS) exhibit strong seasonal and interannual variability that are dominated by changes in soil moisture storage (ΔSMS). Statistical analyses reveal low to moderate correlations (r: -0.30 to -0.56) between ΔTWS, ΔSMS, ΔGWS and the El
Niño-Southern Oscillation (ENSO), particularly during the 2015-2016 El Niño when ΔTWS declined at a rate of 3.8 cm/month. Downscaled ΔTWS (0.25° and 0.5°) are strongly correlated (r = 0.85 to 1) to ΔTWS at coarser spatial scales (3° mascon and the entire Borneo) despite GRACE's native spatial resolution limitations. As a residual parameter, GRACE ΔGWS is subject to arithmetic uncertainties that arise primarily from uncertainty in GRACE products and simulated storage components. Across the 36 realizations employed in this study, ~30% of the GRACE-derived ΔGWS estimates per realization are physically
implausible, exhibiting positive values during dry periods and vice versa; main sources of uncertainty derive from estimates of ΔSMS and surface water storage anomalies (ΔSWS) in this tropical, data-sparse environment. Despite these limitations, plausible GRACE ΔGWS values generally align with groundwater-level dynamics and trends observed from available piezometric data. High-frequency (hourly) groundwater-level observations indicate that episodic, high-intensity rainfall events (>90th percentile) disproportionately contribute to groundwater recharge.

## 1 Introduction

Reliable freshwater resources are essential for sustaining human life, supporting aquatic ecosystems, agricultural productivity, and economic stability (Kundzewicz, 2007; Koehler, 2008; Pimentel et al., 1997; Chakravorty and Zilberman, 2000; Pradinaud et al., 2019; Wilson and Carpenter, 1999). Groundwater serves as a crucial climate-resilient water source (Cuthbert et al., 2019a; Taylor et al., 2013b), storing ~24 million $km^3$ of global water reserves to a depth of 2 km (Gleeson et al., 2016; Ferguson
et al., 2021). It supplies about one-third of the world's agricultural irrigation and drinking water needs (Müller Schmied et al.,



2021). Excessive groundwater abstraction has led to global-scale groundwater depletion (Jasechko et al., 2024; Wada et al., 2010; Konikow and Kendy, 2005; Bierkens and Wada, 2019).

Understanding groundwater storage responses to climate variability and change helps to inform the resilience of groundwater withdrawals to meet increasing freshwater demands (Taylor et al., 2013b; Cuthbert et al., 2019a; Loaiciga and Doh, 2024; Wada et al., 2011; Ferguson and Gleeson, 2012). Such understanding is constrained in the Lower Kutai Basin (LKB) of Indonesia where the country's new capital (Nusantara) is under development, by a very limited, ground-based monitoring network. Freshwater demand in Nusantara is projected to increase at least fourfold for domestic use alone by 2045 (Susantono, 2022). Seasonal imbalances between supply and demand along with potential construction delays can lead to intensified groundwater abstraction for domestic, industrial, and agricultural water use through the drilling of private wells (Grönwall and Danert, 2020). Assessing historical groundwater storage (GWS) changes in response to pumping and recharge is required to inform renewable groundwater use within the basin (Cuthbert et al., 2023).

Recent advancements in satellite-based remote sensing have enabled the monitoring of groundwater storage (GWS) changes using Gravity Recovery and Climate Experiment (GRACE) datasets across diverse hydrological settings at both global (Ndehedehe et al., 2023; Shamsudduha and Taylor, 2020; Thomas et al., 2017; Li et al., 2019; Jin and Feng, 2013; Forootan et al., 2024; Rodell et al., 2024) and basin scales (Rodell et al., 2009; Asoka et al., 2017; Zhang et al., 2024; Rateb et al., 2020; Thomas and Famiglietti, 2019; Ouma et al., 2015). GRACE detects mass variations by measuring changes in Earth's gravity field, which, once corrected for atmospheric and oceanic effects, primarily reflect changes in terrestrial water storage ($\Delta$TWS) (Landerer and Swenson, 2012). $\Delta$TWS encompasses changes in soil moisture, surface water, snow, groundwater, and vegetation water content (canopy storage). Accounting for changes in soil moisture, surface water, snow and canopy storage using global-scale (e.g. Global Land Data Assimilation System, GLDAS) simulations, GRACE enables the indirect estimation of changes in groundwater storage ($\Delta$GWS) as a residual component (Zhang et al., 2024; Shamsudduha and Taylor, 2020; Thomas et al., 2017; Ouma et al., 2015; Rodell et al., 2009).

Climate variability is widely recognized as a key driver of $\Delta$TWS variability (Scanlon et al., 2022; Rodell et al., 2024; Scanlon et al., 2023; Bolaños et al., 2021; Thomas and Famiglietti, 2019; Ni et al., 2018). Among the most pervasive large-scale controls on climate variability is the El Niño-Southern Oscillation (ENSO), a coupled ocean-atmosphere phenomenon originating in the equatorial Pacific Ocean (Trenberth, 1997). ENSO influences precipitation patterns, evapotranspiration rates, and atmospheric circulation (Gu and Adler, 2019; Xu et al., 2004; Tamaddun et al., 2019; Sabziparvar et al., 2011; Moura et al., 2019; Ruiz-Vásquez et al., 2024), thereby modulating water storage dynamics. ENSO events are broadly classified into El Niño (warm phase) and La Niña (cool phase), each of which alters hydrological conditions worldwide. El Niño events typically induce drier-than-normal conditions in regions such as Australia, southern Africa, central China, and Southeast Asia while increasing rainfall in parts of South America, southern China, and East Africa (Wang et al., 2014; Generoso et al., 2020; Kovats, 2000). Conversely, La Niña episodes generally produce the opposite effects in these regions.

Large-scale climatic teleconnections can manifest as fluctuations in $\Delta$TWS and $\Delta$GWS. Studies investigating the relationship between climate indices and $\Delta$TWS anomalies using GRACE data have reported moderate to high correlations



(Ni et al., 2018; Phillips et al., 2012; Scanlon et al., 2022; Bolaños et al., 2021; Anyah et al., 2018; Pereira et al., 2024). For instance, strong regional correlations between GRACE ΔTWS and ENSO have been observed in West and East Africa, Venezuela/Colombia, and Borneo (Phillips et al., 2012; Anyah et al., 2018). Additionally, several studies have demonstrated a relationship between GRACE-derived ΔGWS and ENSO events (Vissa et al., 2019; Kolusu et al., 2019; Forootan et al., 2024; Song et al., 2024), highlighting the role of ENSO in driving interannual groundwater storage variability.

The applicability of GRACE data to assess climate-related ΔTWS and ΔGWS anomalies at spatial scales smaller than the native GRACE resolution of ≥90,000 km² (Loomis et al., 2021; Tapley et al., 2004; Shamsudduha and Taylor, 2020; Wiese et al., 2016) remains poorly characterized. Although Level-3 GRACE datasets are publicly available at higher spatial resolutions such as GRACE CSR at 0.25° and GRACE GSFC at 0.5°, these are resampled from the original coarse-resolution data (Save et al., 2016). Consequently, individual grids are spatially correlated and do not contain independent ΔTWS signals

at scales smaller than GRACE's native resolution (Vishwakarma et al., 2021). Various downscaling techniques have been increasingly applied to estimate ΔTWS and GRACE-derived ΔGWS at finer spatial resolutions. These approaches range from statistical modelling to data-assimilation frameworks incorporating machine learning algorithms (Vishwakarma et al., 2021; Miro and Famiglietti, 2018; Zhong et al., 2021; Fatolazadeh et al., 2022; Yin et al., 2018; Kalu et al., 2024; Verma and Katpatal, 2020; Yin et al., 2022; Yazdian et al., 2023). Although these methods offer practical insights in estimating GRACE-derived

ΔGWS, most remain insufficiently validated. The lack of ground-based observations introduces substantial uncertainty in evaluating the reliability of downscaled GRACE products, particularly for assessing groundwater dynamics at sub-basin scales. Moreover, GRACE ΔTWS estimates are susceptible to signal leakage due to filtering processes, a challenge particularly pronounced in small catchments (Vishwakarma et al., 2016; Vishwakarma et al., 2018). Hydrological models are often used to reduce leakage effects (Landerer and Swenson, 2012; Wiese et al., 2016; Longuevergne et al., 2010); however, model-based

corrections can propagate errors and uncertainties (Vishwakarma et al., 2016).

This study evaluates the applicability of GRACE satellite data to investigate changes in terrestrial water storage (ΔTWS) and groundwater storage (ΔGWS) over the last two decades in the LKB—a coastal, data-scarce region at a scale that is below GRACE's native spatial resolution. In addition to GRACE-derived ΔTWS, we apply data from global-scale models including GLDAS and Water Global Assessment and Prognosis (WaterGAP) models (Rodell et al., 2004; Müller Schmied et al., 2024),

precipitation records from local meteorological stations, and in situ piezometric data from a limited number of monitoring sites. We further examine the influence of large-scale climate variability on water storage dynamics within the basin and assess the plausibility of GRACE-derived ΔGWS estimates. This study contributes to ongoing efforts to improve understanding of the relationship between ΔGWS and climate variability, aiming to support the development of climate-resilient groundwater management strategies, particularly in rapidly urbanizing regions such as Nusantara where water security is a growing concern.




## 2 Materials and methods

### 2.1 Study area

The study area is located within the Lower Kutai Basin (LKB) in East Kalimantan, Indonesia (Fig. 1). It is covered by eight 0.5° grids, which can be further subdivided into smaller 0.25° grids. This region spans approximately 23,175 km² and encompasses diverse landscapes including coastal lowlands, the Mahakam Delta, and hilly uplands with elevations ranging from below 50 m above sea level (masl) to 750 masl.

**Figure 1: (a) Study area within the Lower Kutai Basin (LKB) of East Kalimantan, Indonesia. (b) Distribution of 0.5° and 0.25° grids over Borneo. (c) Elevation data (BIG, 2022), maximum surface water extent (Pekel et al., 2016), and grid distribution across the study area. Green 0.5° grids indicate the selected grids for the study area. Country boundaries are from ESRI (2022) and country abbreviations (e.g., TH, SG, MY) follow ISO codes.**





The LKB is primarily drained by the River Mahakam, the largest river system in East Kalimantan which flows eastward

into the Makassar Strait, forming an extensive deltaic system. The river provides ~70% of the freshwater supply for Samarinda (BPS-Statistics of Kalimantan Timur Province, 2022), the capital city of East Kalimantan Province, whereas the remaining 30% comes from groundwater. In Balikpapan City located south of Nusantara, groundwater serves as the primary water source supplying ~70% of the total water demand (Irsyadulhaq et al., 2024).

The regional hydrostratigraphy is primarily composed of Miocene to Quaternary deltaic deposits which are extensively

distributed across the coastal LKB (KESDM, 2022; Moss and Chambers, 1999). These deposits mainly consist of interbedded sand and clay layers, forming a complex aquifer system. The primary aquifers are sand-dominated sequences extending to depths of ~250 m. These aquifers exhibit significant variability in hydrogeological properties, ranging from low-productivity zones with substantial clay content to highly productive horizons where coarse-grained sands are prevalent (KESDM, 2022). Arifin et al. (2024) provide surface geological and hydrogeological maps of the coastal LKB.

**2.2 GRACE data**

Monthly terrestrial water storage anomalies (ΔTWS) data from 2002 to 2023 were obtained from three Level-3 GRACE datasets: the GRACE JPL RL06.3Mv04 (Landerer et al., 2020; Watkins et al., 2015; Wiese et al., 2023; Wiese et al., 2016) sourced from the NASA PO.DAAC portal (https://podaac.jpl.nasa.gov/); the GRACE CSR RL06.3 (Save et al., 2016; Save, 2020) available from the Center for Space Research (CSR) at the University of Texas at Austin portal

(https://www2.csr.utexas.edu/grace/); and the GRACE GSFC RL06v2.0 (Loomis et al., 2019) provided by the NASA Goddard Space Flight Center (GSFC) via the NASA GSFC portal (https://earth.gsfc.nasa.gov/geo/data/grace-mascons); these datasets are derived from GRACE and its successor, GRACE Follow-On (FO). The recently released Level-3 GRACE data employ mass concentration blocks (mascons) to compute Earth's gravitational field variations resulting from water mass changes. The mascon solution offers several advantages over the conventional spherical harmonic approach, particularly in noise reduction

and the application of geophysical constraints, leading to reduced leakage errors compared to the harmonic solution (Watkins et al., 2015; Scanlon et al., 2016).

The downscaled GRACE JPL and GRACE GSFC datasets provide submascon fields on a 0.5° grid, whereas the downscaled GRACE CSR dataset offers a finer 0.25° grid resolution. Although the downscaled Level-3 GRACE products are publicly available at higher resolutions, the true effective resolution of the data remains coarse due to the inherent limitations

of GRACE's observational design and constraints imposed by processing methods and spatial filtering techniques (Vishwakarma et al., 2021). GRACE JPL applies 3° spherical cap smoothing, whereas GRACE GSFC and CSR use 1° equal-area geodesic grids (Table S1). For GRACE JPL, gain factors are provided separately to account for water mass changes at the 0.5° scale, excluding regions dominated by ice sheets or mountain glaciers. These gain factors, derived from the hydrological components of the Community Land Model (CLM), adjust for sub-mascon-scale variations and can be applied

to land-based water storage signals on the 0.5° grid (Wiese et al., 2016). In contrast, the GRACE CSR and GSFC datasets



resample the coarser data into finer grids (Save et al., 2016; Loomis et al., 2019). It is important to note that both GRACE JPL and CSR datasets caution against using GRACE ΔTWS data for single-grid-based analyses as neighbouring grids are not independent of each other.

The GRACE datasets provide mass change estimates relative to a baseline mean. GRACE JPL and CSR reference a 2004-2009 mean, whereas GRACE GSFC uses a 2004-2010 mean as its baseline. The GRACE CSR and GSFC datasets can be used as-is, as both incorporate necessary corrections for each grid and do not require additional adjustments with finer-scale gain factors. In contrast, GRACE JPL requires users to apply gain factors for the 0.5° grids. Additionally, the GRACE JPL dataset provides conservative uncertainty estimates, but these estimates are only valid at the 3° scale. A notable methodological difference is that the GRACE CSR and GSFC datasets do not include a Coastline Resolution Improvement (CRI) filter, whereas

GRACE JPL incorporates CRI filtering to mitigate leakage errors in coastal regions (Wiese et al., 2016).

## 2.3 GLDAS and WGHM data

This study employs two GLDAS datasets spanning 2003-2023: the monthly Noah Land Surface Model (LSM) L4 V2.1 (Beaudoing and Rodell, 2020; Rodell et al., 2004) and the daily Catchment LSM L4 V2.2 (Li et al., 2020; Li et al., 2019). Both datasets have a 0.25° × 0.25° spatial resolution and are accessible via NASA's Land Data Assimilation Systems (LDAS)

portal (https://ldas.gsfc.nasa.gov/). In contrast, the Variable Infiltration Capacity (VIC) LSM operates at a coarser 1° × 1° resolution and lacks complete coverage for certain regions within the study area.

The GLDAS datasets integrate satellite and ground-based observations with advanced land surface models to produce globally distributed, high-resolution simulations of land surface states and fluxes, including soil moisture, snow water equivalent, canopy water, and surface runoff (Rodell et al., 2004). In this study, soil moisture and canopy water components

are derived from Noah and Catchment LSMs, whereas surface water storage is obtained from the WaterGAP Hydrological Model (WGHM) v2.2e (Müller Schmied et al., 2024) and Noah LSM. Although surface runoff from GLDAS can serve as a proxy for surface water storage (Shamsudduha and Taylor, 2020; Zhang et al., 2024; Ali et al., 2021), runoff data from the Catchment LSM are excluded due to their implausible magnitudes (Fig. S1). In contrast, both WGHM and Noah LSM produce more reasonable ΔSWS estimates. Furthermore, WGHM includes storage in rivers, lakes, reservoirs, and wetlands, offering a

more comprehensive representation of surface water. Snow water equivalent is excluded from water storage calculations due to the tropical climate of East Kalimantan, where snow water contributions are zero.

To incorporate GLDAS and WGHM data with GRACE for groundwater storage change calculations, it is essential to maintain consistency in parameter units (cm), temporal resolution (monthly), and the baseline period (2004-2009). Since GRACE observations are available at a monthly resolution, the daily Catchment LSM dataset is aggregated to a monthly

timescale to align with GRACE data. Additionally, both GLDAS and WGHM datasets are adjusted to the 2004-2009 GRACE baseline mean, ensuring that water storage anomalies are computed relative to a consistent reference period.



### 2.4 Computation of groundwater storage changes

Groundwater storage changes (ΔGWS) are estimated by integrating monthly terrestrial water storage anomalies (ΔTWS) from GRACE satellite data with simulated hydrological components from GLDAS and WGHM datasets. The simulated variables include soil moisture storage anomalies (ΔSMS), plant canopy water anomalies (ΔCW), and surface water storage anomalies (ΔSWS). This approach enables an indirect estimation of groundwater storage variations by isolating the residual water storage component that is not accounted for by surface, soil, or vegetation storage components.

The calculation of ΔGWS follows standard methodologies outlined in previous global-scale (e.g. Thomas et al., 2017; Shamsudduha and Taylor, 2020; Ndehedehe et al., 2023) and basin-scale studies (e.g. Rodell et al., 2009; Asoka et al., 2017; Zhang et al., 2024; Rateb et al., 2020; Thomas and Famiglietti, 2019). Given the tropical climate of the study area, snow water equivalent (SWE) changes are excluded from the computation. The relationship between these variables is expressed as:

$$\Delta GWS = \Delta TWS - (\Delta SMS + \Delta SWS + \Delta CW) \tag{1}$$

By subtracting the combined contributions of ΔSMS, ΔSWS, and ΔCW from ΔTWS, ΔGWS isolates groundwater storage variations, capturing the residual component of the terrestrial water budget that is primarily stored in aquifers.

### 2.5 Precipitation, evapotranspiration, and piezometry data

Rainfall monitoring in the study area is available at two key meteorological stations: Samarinda and Balikpapan. The region has a tropical monsoonal climate with high annual precipitation that exhibits substantial interannual variability. Over the period 2000-2023, annual rainfall ranged from 1,420 mm to 3,960 mm (Fig. S2). Analysis of monthly precipitation boxplots from both stations (Fig. S3) reveals distinct seasonal precipitation patterns in which September (median: 168 mm) is the driest month in Balikpapan and August (median: 101 mm) is the driest month in Samarinda. Conversely, the wettest month is in December in Balikpapan (median: 269 mm) and April is the wettest month in Samarinda (median: 274 mm).

To investigate further precipitation trends, a non-parametric smoothing approach, Locally Estimated Scatterplot Smoothing (LOESS) (Cleveland, 1979), is applied to precipitation data from both stations (Fig. S4 and S5). The LOESS-generated smoother curves provide insights into interannual and seasonal precipitation patterns, highlighting potential shifts in rainfall seasonality and long-term trends. Extreme monthly precipitation events are defined statistically as those exceeding the 90th percentile. This study also incorporates evapotranspiration data from the monthly Noah LSM dataset to compute the standardized precipitation evapotranspiration index (SPEI), a climate index used to characterize meteorological drought conditions, integrating both precipitation and potential evapotranspiration (Vicente-Serrano et al., 2010). The SPEI is calculated for 1-, 3-, and 6-month periods to capture short- and medium-term drought variability (Fig. S4 and S5).

Groundwater level (GWL) monitoring in the study area is constrained to the southern part of Balikpapan City where piezometric data are available from just five sites, comprising four monitoring wells and one pumping well (Fig. 1 and S6).



For the pumping well, only the shallowest daily GWL values during the recovery phase are employed to minimize the influence
of transient drawdown effects on long-term groundwater trend assessments. Groundwater level depths are reported in metres
below ground level (mbgl).

## 2.6 Climate indices

This study employs four climate indices (Fig. S7): two El Niño-Southern Oscillation (ENSO) indicators, one Indian Ocean
Dipole (IOD) index, and one Pacific Decadal Oscillation (PDO) index. ENSO is a climatic phenomenon characterized by
periodic fluctuations in sea surface temperatures (SST) and atmospheric conditions over the equatorial Pacific Ocean
(Trenberth, 1997). IOD is defined by SST anomalies in the Indian Ocean (Saji et al., 1999), whereas the PDO represents long-
term SST fluctuations predominantly observed in the North Pacific Ocean (Mantua and Hare, 2002; Mantua et al., 1997).

The two ENSO indicators used in this study are the Multivariate ENSO Index (MEI) and the Oceanic Niño Index (ONI).
MEI is one of the most comprehensive tools for assessing ENSO conditions, as it integrates multiple meteorological and
oceanographic variables, including sea-level pressure, zonal and meridional components of surface wind, sea surface
temperature, surface air temperature, and total cloudiness fraction (Wolter and Timlin, 1998; Wolter and Timlin, 2011). MEI
values greater than 0.5 indicate El Niño conditions whereas values lower than -0.5 indicate La Niña conditions (Kiem and
Franks, 2001). The ONI, on the other hand, is closely related to the Niño 3.4 index which focuses specifically on SST anomalies
within the central equatorial Pacific (5°N-5°S, 170°W-120°W) (Bamston et al., 1997). The ONI employs a 3-month running
mean of SST anomalies; the National Oceanic and Atmospheric Administration (NOAA) defines El Niño or La Niña events
as occurring when these anomalies surpass ±0.5°C for at least five consecutive months.

The Dipole Mode Index (DMI) is the standard metric used to quantify the IOD, calculated as the difference in SST
anomalies between the western (50°E-70°E, 10°S-10°N) and southeastern (90°E-110°E, 10°S-0°N) tropical Indian Ocean
regions (Saji et al., 1999). The IOD oscillates between two phases. The positive phase marked by warmer SST in the western
Indian Ocean and cooler SST in the eastern part; the negative phase exhibits the opposite pattern. The PDO index exhibits
multi-decadal phases that can last 20-30 years (Mantua and Hare, 2002). It is characterized by SST shifts in the North Pacific,
where the positive (warm) phase features warmer SST along the North American west coast and cooler SST in the central
North Pacific. The negative (cool) phase exhibits the opposite pattern. The climate indices used in this study were retrieved
from NOAA Physical Sciences Laboratory (PSL) portal (https://psl.noaa.gov/).

## 2.7 Seasonal and trend decomposition

This study applies Seasonal and Trend decomposition using LOESS (STL), a robust non-parametric method introduced by
Cleveland et al. (1990), to decompose a time series dataset into three distinct components: trend, seasonal, and residual. The
trend component represents long-term increasing or decreasing patterns, whereas the seasonal component captures recurring
periodic variations that typically follow a seasonal cycle. The residual component accounts for irregular fluctuations that do
not conform to a predictable structure.



The LOESS-based smoothing approach used in STL is particularly versatile, as it does not assume any predefined structure in the data. This flexibility allows for the identification of complex, non-linear trends that might otherwise be overlooked by traditional methods such as linear regression (Jacoby, 2000). STL has been widely adopted in hydrological and climate studies, including those involving GRACE data (e.g. Hassan and Jin, 2014; Humphrey et al., 2016; Jing et al., 2019;

Rateb et al., 2020; Liesch and Ohmer, 2016; Shamsudduha and Taylor, 2020; Ouma et al., 2015; Ali et al., 2024).

For example, the STL decomposition process for groundwater storage changes (ΔGWS) can be expressed as:

$$\Delta GWS_t = T_t + S_t + R_t \tag{2}$$

where $\Delta GWS_t$ represents the observed $\Delta GWS$ at time $t$, and $T_t$, $S_t$, and $R_t$ denote the trend, seasonal, and residual components, respectively. Since STL requires evenly spaced time series data, linear interpolation is applied to fill gaps arising from missing GRACE observations.

STL decomposition is achieved through successive smoothing operations that extract different frequencies from the time series. Selecting appropriate smoothing parameters for both the trend and seasonal components is critical. Shamsudduha and

Taylor (2020) experimented with various window widths and found that a window width of 13 for the seasonal component and 37 for the trend component effectively captured the structure of time series datasets in 37 major global aquifer systems. These values are also adopted in this study.

In addition to STL decomposition, this study applies the Theil-Sen estimator to compute trends of time series data. The Theil-Sen method is a non-parametric regression technique that estimates the median slope of a dataset, making it more robust than simple linear regression as it is less sensitive to outliers and data gaps (Sen, 1968). To assess the statistical significance

of detected trends, this study employs the Mann-Kendall test, a widely used rank-based non-parametric statistical test for detecting monotonic trends in hydrological and climatological time series (Mann, 1945).

## 3 Results

### 3.1 Comparative analysis of mean ΔTWS across GRACE datasets

Mean terrestrial water storage anomaly (ΔTWS) values from three GRACE datasets (JPL, GSFC, and CSR) were statistically analyzed and compared (Tables S2 and S3). To evaluate the applicability of using GRACE-derived ΔTWS for a study area smaller than the native GRACE resolution of $\geq 90{,}000$ km² (Loomis et al., 2021; Tapley et al., 2004; Shamsudduha and Taylor, 2020; Wiese et al., 2016), the ΔTWS values were classified into three categories based on spatial (grid) scale: (1) the study area grids consisting of an ensemble of eight 0.5° grids for GRACE JPL and GSFC and the corresponding 0.25° grids for

GRACE CSR; (2) a single 3° mascon unit (as derived from GRACE JPL); and (3) the entire Borneo Island (Fig. 1b). These classifications allow for a comparative assessment of ΔTWS variability across different spatial scales and provide insights into how well GRACE data capture hydrological signals in smaller regions.



For each GRACE dataset, the ΔTWS of the study area exhibits strong similarity to both the ΔTWS of a single 3° mascon and that of the entire Borneo Island (Fig. 2). A comparative analysis of mean ΔTWS values (Table S2) reveals strong

correlations (r = 0.85 to 1) and relatively low root mean square errors (RMSE: 0.8 to 4.4 cm). The mean ΔTWS of the study area closely follows the ΔTWS of the single 3° mascon across all GRACE datasets, with r values ranging from 0.96 to 1 and RMSE values between 0.8 and 2.5 cm.

**Figure 2: Comparison of mean ΔTWS values from GRACE datasets: (a) JPL, (b) GSFC, and (c) CSR. In (a) to (c), the black, orange, and blue lines represent ΔTWS for the study area, a single 3° mascon, and the entire Borneo, respectively. Plot (d) shows the mean ensemble ΔTWS of the study area (red line) derived from the three GRACE datasets (gray lines). Missing ΔTWS data are indicated by gaps in the dataset.**





These measures suggest that, despite GRACE's inherent spatial limitations, ΔTWS signals in the study area are consistent with those observed ΔTWS at the mascon scale. ΔTWS values for the study area diverge slightly from those of Borneo, particularly in the GRACE GSFC dataset which exhibits the lowest correlation (r = 0.85) and the highest RMSE (4.4 cm). The magnitude of within-year reductions in ΔTWS also varies by spatial scale, with Borneo exhibiting the smallest ΔTWS reductions relative to the study area, likely due to regional averaging that dampens localized hydrological variability.

Statistical differences (Table S3) among the datasets highlight dataset-specific characteristics in ΔTWS representation. Among all ΔTWS values from the three GRACE datasets, GRACE JPL (skewness: -0.34 to -0.07) displays less extreme negative anomalies compared to GRACE GSFC (skewness: -1.03 to -0.31) and GRACE CSR (skewness: 0.97 to -0.45). GRACE GSFC exhibits the most extreme negative ΔTWS (-23.7 cm) compared to GRACE CSR (-15.5 cm) and GRACE JPL (-12.9 cm). ΔTWS values from GRACE JPL, in contrast, show the highest positive signals (19.3 cm), followed by GRACE
GSFC (15.5 cm) and GRACE CSR (14.6 cm). This study employs ΔTWS from each GRACE dataset as well as the mean ensemble (Fig. 2d) to compute groundwater storage changes (ΔGWS).

## 3.2 Simulated water storage components from GLDAS and WGHM

In the study area, water storage components derived from the GLDAS and WGHM datasets exhibit distinct contributions to mean ensemble ΔTWS. Among these components, soil moisture storage changes (ΔSMS) play the most dominant role with
values ranging from -28.8 to 5.5 cm (Table S4 and Fig. S8). In contrast, surface water storage changes (ΔSWS) derived from the Noah LSM and WGHM datasets are comparatively smaller, ranging from -3.7 to 4.1 cm (Table S4, Fig. S1). Canopy water anomalies (ΔCW) are negligible with values ranging from -0.02 to 0.01 cm (Table S4, Fig. S9). The very low ΔCW values indicate that canopy water variations have an insignificant influence on overall water storage changes, whereas small ΔSWS values suggest that surface water contributes minimally to ΔTWS in the study area.

Between the two GLDAS datasets, ΔSMS from Noah LSM exhibits greater variability and stronger negative skewness (-1.93) than Catchment LSM (-1.06). The pronounced skewness in Noah LSM ΔSMS indicates an asymmetric distribution with a higher frequency of extremely low values, suggesting that the Noah LSM is more responsive to extreme moisture deficits. Conversely, the skewness in the Catchment LSM reflects a more evenly distributed dataset with fewer extreme negative values, indicating a more stable soil moisture regime. These differences highlight the sensitivity of Noah LSM to
extreme hydrological conditions, which may be advantageous for detecting short-term anomalies but could also introduce greater uncertainty when assessing long-term trends.

Despite these dataset-specific variations, ΔSMS values from Noah and Catchment LSMs exhibit a strong positive correlation (r = 0.86), indicating that both datasets capture similar trends in soil moisture variations over the study area. This high correlation suggests that although ΔSMS values may differ between models due to variations in parameterization and
calibration, the overall patterns of soil moisture fluctuations remain consistent across both datasets. This study employs ΔSMS values from both Noah and Catchment LSMs as well as their mean ensemble to compute groundwater storage changes




(ΔGWS). For ΔCW, the mean ensemble of values derived from the two models is used. ΔSWS is derived from the WGHM and Noah LSM datasets. The substantial temporal variability in surface water extent from global-scale estimates (Pekel et al., 2016) further underscores the important contribution of surface water to ΔTWS (Fig. S1d).

### 3.3 ΔTWS and ΔSMS variability

Mean ensemble ΔTWS derived from the three GRACE datasets (JPL, GSFC, and CSR) exhibits substantial interannual variability from 2002 to 2023 (Fig. 2d). These fluctuations are characterized by alternating positive anomalies indicating increases in water storage, and negative anomalies signifying periods of water storage decline. The mean ensemble ΔTWS ranges from -15.9 to 12.9 cm.

Analysis of GRACE-derived annual linear ΔTWS trends within the study area provides insights into long-term hydrological changes. The 3° GRACE JPL mascon dataset reveals an annual ΔTWS trend of 0.6 cm/year at the one-mascon scale (Fig. 3a and 3e). At finer spatial resolutions, mean annual linear ΔTWS trends are of lower magnitudes: 0.5 cm/year for GRACE JPL (0.5° grids), 0.31 cm/year for GRACE GSFC (0.5° grids), and 0.27 cm/year for GRACE CSR (0.25° grids). All trends are statistically significant with p-values < 0.05. Closer to the native resolution of GRACE, the 3° GRACE JPL mascon data reveal the strongest positive trend of 1.4 cm/year in the Sarawak region of Malaysia (Fig. 3a). This trend slightly decreases to 1.36 cm/year in the 0.5° ensemble of GRACE JPL, increases to 1.6 cm/year in the 0.5° ensemble of GRACE GSFC, and is substantially lower at 0.5 cm/year in the 0.25° ensemble of GRACE CSR. Additionally, ΔTWS trend signals of each 0.5° grids within the study area (Fig. 3e) vary considerably across GRACE datasets, reinforcing the recommendation to avoid single-grid-based analyses and instead employ ensemble mean values for more robust interpretations (Vishwakarma, 2020).

The spatial distribution of ΔTWS signals across different grids within the study area reveals dataset-specific variability. The GSFC and CSR datasets exhibit consistent ΔTWS patterns across all grids, whereas GRACE JPL shows localized discrepancies in two of the eight grids (Fig. S10). These two grids remain relatively stable over time, showing minimal ΔTWS variations. Gain factor analysis of the GRACE JPL dataset (Fig. S11) further highlights these differences: two grids within the JPL dataset have negative gain factors (-0.07), whereas the remaining six grids exhibit positive gain factors (0.83 and 1.4). Positive gain factors enhance ΔTWS variability, leading to ΔTWS patterns that closely match those from the GRACE GSFC and CSR datasets. Conversely, negative gain factors, particularly in the Mahakam Delta area, suppress ΔTWS fluctuations, resulting in significantly lower ΔTWS variability, ranging only from -1.3 to 0.9 cm from 2002 to 2023 period. The dampening effect of negative gain factors suggests potential data filtering artifacts which may reduce GRACE sensitivity.

To investigate water storage variability, seasonal-trend decomposition using LOESS (STL) analysis is applied to mean ensemble ΔTWS values, revealing a nonlinear trend (Fig. S12). Among the decomposed components, the residual component accounts for the highest ΔTWS variance (14.6 cm²), followed by seasonal (9 cm²) and trend (7.1 cm²) components. The dominance of the residual component suggests that short-term irregular ΔTWS fluctuations drive significant hydrological variability in the study area.






**Figure 3: Comparison of GRACE datasets: (a) JPL (3° mascon grids), (b) JPL (0.5° grids), (c) GSFC (0.5° grids), and (d) CSR (0.25° grids). The maps display the annual linear trend of terrestrial water storage anomalies (ΔTWS) derived using the Theil-Sen method. Black dots indicate areas where p-values are < 0.05, signifying statistical significance. Plot (e) displays the annual linear ΔTWS trends for each grid within the study area across all GRACE datasets.**



A similar pattern emerges when analyzing ΔSMS from GLDAS over the 2003-2023 period. Like ΔTWS, ΔSMS exhibits alternating positive and negative anomalies, representing soil moisture increases and deficits. STL analysis of mean ensemble ΔSMS values also reveals a nonlinear trend (Fig. S13), with the residual component accounting for the highest variance (13.7 cm²), followed by seasonal (5.1 cm²) and trend (1.1 cm²) components. These results confirm that soil moisture fluctuations are the primary driver of short-term terrestrial water storage variability in the study area, reinforcing the dominant role of ΔSMS

in controlling overall ΔTWS dynamics.

### 3.4 ΔGWS estimates

As a residual parameter (eq. 1), ΔGWS is subject to propagated errors and uncertainties from filtered GRACE ΔTWS and global-scale models used in its computation (Vishwakarma et al., 2016), which can lead to arithmetic problems as pointed out by Shamsudduha and Taylor (2020). Simulations from global-scale models have limitations in their ability to represent

accurately changes in individual terrestrial stores (e.g. ΔSMS, ΔSWS) as highlighted, for example, by their inferred collective comparison with GRACE ΔTWS (Scanlon et al., 2018). Discrepancies in individual stores (e.g. ΔSMS) can directly influence computation of GRACE  ΔGWS from GRACE ΔTWS. Fig. 4 shows that when ΔSMS values from Noah LSM are more negative than ΔTWS values from GRACE JPL (0.5°), GRACE ΔGWS becomes anomalously high and vice versa. For example (Table 1), during the wet season, ΔTWS in March 2008 was 1.9 cm, whereas ΔSMS was 5 cm and ΔSWS was 2.8 cm. The

computed ΔGWS of -5.8 cm is implausible unless intense and widespread groundwater abstraction has occurred. Conversely, in September 2015 when precipitation was zero, ΔTWS was -8.6 cm, ΔSMS was -22.5 cm, and ΔSWS was -3.2 cm. Despite the overall water deficit, the computed ΔGWS was 6.2 cm, an implausible gain unless significant regional groundwater inflow into the GRACE grid occurred. These examples highlight computational uncertainty in ΔGWS estimation, particularly when specific dataset combinations are used.


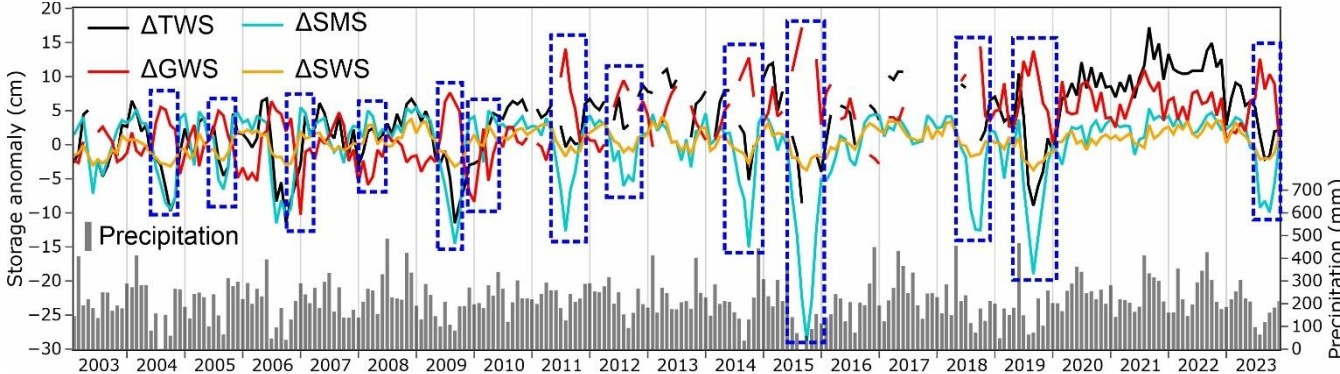

**Figure 4: Comparison between ΔTWS (GRACE JPL 0.5°), ΔSMS (Noah LSM), ΔSWS (WGHM), and computed ΔGWS. Blue dashed polygons indicate arithmetic problems resulting from ΔSMS values being significantly lower than ΔTWS.**




**Table 1: Examples of arithmetically implausible ΔGWS estimates based on the comparison between ΔTWS (GRACE JPL 0.5°), ΔSMS (Noah LSM), and ΔSWS (WGHM).**

| Month-year | Precipitation (mm) | ΔTWS (cm) | ΔSMS (cm) | ΔSWS (cm) | ΔGWS (cm) |
|---|---|---|---|---|---|
| March-2008 | 265 | 1.9 | 5 | 2.8 | -5.8 |
| Sep-2009 | 81 | -11.5 | -14.5 | -3.2 | 6.2 |
| April-2010 | 282 | 2.5 | 5 | 2.8 | -5.3 |
| Sep-2015 | 0 | -8.6 | -22.5 | -3.2 | 17.1 |
| Sep-2019 | 73 | -8.9 | -19 | -3.7 | 13.8 |

This study employs 36 realizations to compute ΔGWS from 2003 to 2023, using four ΔTWS datasets (JPL, GSFC, CSR, and the mean ensemble), three ΔSMS datasets (Noah LSM, Catchment LSM, and the mean ensemble), three ΔSWS datasets (WGHM, Noah LSM, and the mean ensemble), and one ΔCW dataset (ensemble mean of Noah and Catchment LSMs). The results indicate that ~30% of GRACE-derived ΔGWS estimates from each realization are implausible (Table S5), primarily due to two conditions: (1) during the wet season, when ΔTWS, ΔSMS, and ΔSWS are all positive, ΔTWS is lower than the combined ΔSMS and ΔSWS, resulting in anomalously negative ΔGWS values; and (2) during the dry season, when all components are negative, the absolute value of ΔTWS is smaller than the sum of the absolute values of ΔSMS and ΔSWS, yielding anomalously positive ΔGWS estimates. The mean ensemble of plausible ΔGWS estimates across realizations yields ~85% of ΔGWS estimates that are suitable for further analysis (Fig. 5).

The mean ensemble of plausible ΔGWS values exhibits substantial variability (Fig. 5), ranging from -5.7 to 8.7 cm, with an annual linear trend of 0.2 cm/year (p-value < 0.05). STL decomposition of the ensemble mean ΔGWS reveals a nonlinear trend (Fig. S14), with the trend component accounting for the greatest variance (3.4 cm²), followed by the residual (3.3 cm²) and seasonal (1.1 cm²) components. The correlation coefficient (r = 0.85) further indicates a strong relationship between mean ensemble ΔTWS and plausible ΔGWS estimates. The inclusion of arithmetically implausible ΔGWS values significantly reduces this correlation to 0.33 (Fig. 6). The mean ensemble ΔTWS also exhibits a strong correlation with mean plausible ΔSMS (r = 0.88), reinforcing that soil moisture is the dominant driver of ΔTWS variability. Plausible ΔSMS and ΔGWS values are employed in analyses presented below in this study.

## 3.5 The influence of climate variability on water storage changes

The relationship between mean ensemble ΔTWS, ΔSMS, and computed ΔGWS is related to four major climate indices (MEI, ONI, DMI, and PDO) from 2003 to 2023 in the study area. Both ΔTWS and ΔSMS exhibit stronger negative correlations with ENSO-related indices (MEI and ONI) than with the IOD (DMI) and the PDO, suggesting ENSO is the dominant climate driver of regional water storage variations. Among all tested relationships, ΔSMS shows the strongest response to ENSO with correlation coefficients (r) of -0.56 with MEI and -0.54 with ONI. ΔTWS also exhibits notable correlations with MEI (r = -0.52) and ONI (r = -0.45).





**Figure 5: Mean ensemble of (a) ΔTWS (black line), (b) ΔSMS (blue line), and (c) ΔSWS (orange line). Plot (d) shows arithmetically plausible mean ensemble ΔGWS (red circles) from 36 realizations (gray circles) and mean precipitation. Dashed lines in (a) to (d) indicate the trend components derived from STL analysis. Gray lines in (a) and (b) represent ΔTWS and ΔSMS from individual GRACE and GLDAS datasets. In (c), gray lines indicate ΔSWS from the Noah LSM and WGHM datasets.**

The weak correlation between ΔTWS and DMI (r = -0.24) and between ΔSMS and DMI (r = -0.28) suggests that IOD-induced changes in regional monsoon patterns and moisture transport mechanisms may contribute to water storage variability to a lesser extent than ENSO. ΔTWS and ΔSMS also exhibit low correlations with PDO, with r values of -0.24 and -0.22, respectively. These statistical relationships imply that longer-term Pacific decadal variability exerts only a minor influence on short-term water storage fluctuations in the study area. Unlike ΔTWS and ΔSMS, the computed mean ensemble ΔGWS shows



consistently weaker correlations with all climate indices. The strongest correlation is observed between ΔGWS and MEI (r = -0.35), whereas the weakest correlation is with DMI (r = -0.17).

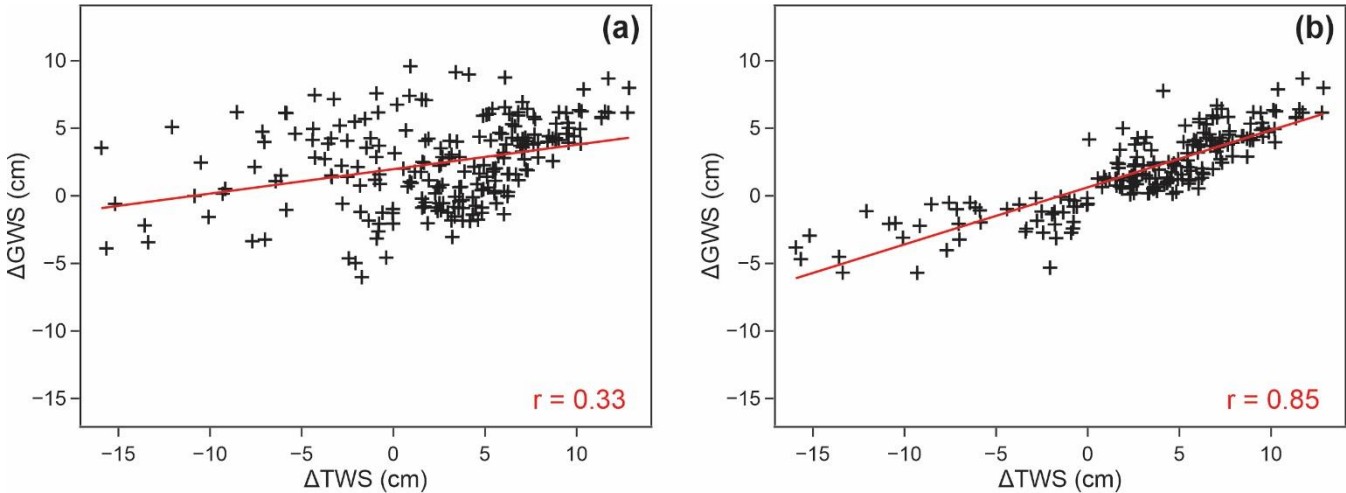

**Figure 6: Cross-correlations between mean ΔTWS and ΔGWS: (a) using all ΔGWS estimates and (b) excluding the physically implausible ΔGWS estimates.**

Although the overall correlations between ENSO indices, ΔTWS, and ΔSMS are moderate (r = -0.4 to -0.6), the impact of strong ENSO events is much more pronounced, particularly during extreme El Niño and La Niña phases. This is indicated by strongly positive or negative ONI values exceeding ±0.5°C for at least five consecutive months, accompanied by corresponding sustained positive or negative MEI values. These strong ENSO events coincide with the most extreme ΔTWS and ΔSMS fluctuations in the study area (Fig. 7). A notable example is the 2015-2016 El Niño, recognized as one of the strongest on record (L'Heureux et al., 2017). This event resulted in record-low ΔTWS and ΔSMS estimates, indicating severe terrestrial water deficits likely driven by widespread drought conditions, as also reflected in strong negative SPEI values (Fig. 7d). The ΔTWS and ΔSMS responses during this extreme El Niño underscore the sensitivity of water storage components to large-scale climatic forcing, particularly ENSO.

## 4 Discussion

### 4.1 Comparison of GRACE-derived ΔGWS with in situ piezometric observations

One of the major limitations in using GRACE-derived ΔGWS is its coarse spatial resolution (≥90,000 km²), which restricts its ability to resolve localized groundwater level fluctuations (Loomis et al., 2021; Tapley et al., 2004; Shamsudduha and Taylor, 2020; Wiese et al., 2016). Consequently, ΔGWS estimates may deviate substantially from in situ piezometric observations, particularly in regions where localized factors such as groundwater abstraction introduce variability (Shamsudduha and Taylor, 2020; Asoka et al., 2017). For example, Asoka et al. (2017) reported that GRACE-derived ΔGWS trends in India



underestimated those derived from piezometric data, likely due to localized depletion not captured at GRACE's coarse resolution. Similarly, Shamsudduha and Taylor (2020) reported increasing GRACE-derived ΔGWS trends in the Guarani Aquifer System (South America) despite observed groundwater depletion caused by intensive urban and agricultural groundwater withdrawals during the same period.




**Figure 7: Comparison of (a) mean ensemble ΔTWS and ΔSMS, (b) computed mean ensemble ΔGWS, (c) MEI and ONI indices, and (d) 3-month SPEI. In (c), red and blue shadings represent El Niño and La Niña episodes, respectively, where ONI exceeds ±0.5°C. The strongest recorded El Niño event (2015/2016) is highlighted by a red dashed polygon.**





In this study, ΔGWS is compared to groundwater level changes (ΔGWL) as the storage coefficient (S or Sy) is not known. ΔGWS values exhibit reasonable agreement (Fig. 8) with ΔGWL trends (r = -0.55 to 0.43). At site MM (screened intervals: 123-126 m), ΔGWL declined in 2020 and increased in the following years, moderately mirroring ΔGWS variations (r = 0.43). Similarly, at sites GM (screened interval: 96-99 m) and TS (screened interval: 70-80 m), ΔGWS weakly follows ΔGWL (r = 0.35) during the 2015-2017 period with both showing a declining trend in 2015-2016 followed by a recovery in 2017-2018,
indicative of subsequent groundwater recharge.

        In several instances, piezometric observations show ΔGWL trends that differ from ΔGWS signals. At site MG (screened interval: 87-135 m), ΔGWL exhibits a continuous decline from 2015 to 2023, with rates of -0.9 m/year (p-value < 0.05) for the 87-90 m interval and -0.6 m/year (p-value <0.05) for the 132-135 m interval. Similarly, at site MM (screened interval: 21-24 m), ΔGWL dropped significantly from 5.1 m to 0.1 m by the end of 2019, followed by a relatively stable trend through
2023. These discrepancies may be attributable to anthropogenic groundwater abstraction (Fig. S15) which may play a crucial role in local groundwater level changes, often overriding regional-scale ΔGWS trends observed from GRACE data.

        The ΔGWS trend of 0.2 cm/year (p-value < 0.05) over the last two decades may indicate an aquifer-full condition in the study area. This interpretation is supported by intra-annual ΔTWS patterns, which display substantial declines during periods of low precipitation but rapid recoveries following increases in rainfall (Fig. 5). Additional evidence is provided by
groundwater level observations at the MG site (screened interval: 30-33 m), where water levels have remained very shallow, fluctuating between 0.3 and 1.4 meters below ground level (mbgl) from 2015 to 2023, with a mean of $1.0 \pm 0.1$ m. Groundwater abstraction is largely concentrated near the southern coast of Balikpapan City (Fig. S15), likely contributing to localized declines in groundwater levels.

## 4.2 ENSO influence on water storage variability

This study identifies moderate correlations between ENSO indices (MEI and ONI) and terrestrial water storage components (ΔTWS and ΔSMS) with r values ranging from -0.45 to -0.56, indicating that ENSO events exert a relatively strong influence on water storage variability in the study area. In contrast, ΔGWS shows a weaker response to ENSO events with r values ranging from -0.24 to -0.35. The correlation between ENSO indices and precipitation is also weak, with MEI and ONI producing r values of -0.33 and -0.30, respectively. Moreover, no significant lag effect is observed between ΔTWS, ΔSMS,
ΔGWS, or precipitation and ENSO events; attempts to introduce 1- to 6-month lags resulted in weaker correlations. For example, the correlation between MEI and ΔTWS declines from -0.48 at a 1-month lag to -0.17 at a 6-month lag, suggesting that water storage anomalies in the study area respond contemporaneously to ENSO variability. This aligns with previous findings which reported no significant lag between ΔTWS and ENSO-related teleconnection indices across most of Indonesia (Liu et al., 2020), which may be consistent with an aquifer-full condition in the study area.






**Figure 8: Comparison between ΔGWL (black, blue, and red circles) at (a) MG, (b) MM, (c) GM, and (d) TS sites with the mean ensemble ΔGWS (green circles-lines), and (e) MEI (green line) and ONI (orange line) indices. Both ΔGWL and ΔGWS are referenced to the 2015-2023 mean baseline. Correlations (r) between ΔGWS and ΔGWL are indicated. In (e), red and blue shadings represent El Niño and La Niña episodes, respectively, where ONI exceeds ±0.5°C.**





Plots of mean ensemble ΔTWS, ΔSMS, and ΔGWS alongside ENSO indices (MEI, ONI) and the 3-month SPEI (Fig. 7) further confirm a strong correlation between water storage anomalies and ENSO events in the study area. Documented El Niño events in 2004, 2006, 2009, 2015/2016, 2018/2019, and 2023 (Kamra and Athira, 2016; McPhaden, 2004; Lee et al., 2020;

Raghuraman et al., 2024) coincide with negative SPEI values, indicative of pronounced dry conditions during these periods. Among these, the 2015-2016 El Niño stands out as one of the most intense El Niño episodes in recent decades, leading to severe drought across Southeast Asia and increased forest fires in Indonesia (Santoso et al., 2017; Kogan and Guo, 2017; L'Heureux et al., 2017).

During the February-October 2015 El Niño period, water storage anomalies in the study area exhibited significant

negative trends. ΔTWS declined at a rate of -3.8 cm/month and ΔSMS exhibited a similar downward trend of -3.4 cm/month. Both trends are statistically significant with p-values < 0.05. The computed ΔGWS also displayed a decreasing trend of -2.1 cm/month, also statistically significant (p-value < 0.05). Similar responses have been observed in other regions. Southern Africa experienced severe drought and water storage depletion during the 2015-2016 El Niño, whereas Eastern Africa experienced anomalously wet conditions, leading to increases in ΔTWS and groundwater recharge as observed in both GRACE

data and in situ piezometric records (Kolusu et al., 2019; Scanlon et al., 2022).

The transition to La Niña phases, such as the 2010 event, one of the strongest La Niña episodes in recent history (Boening et al., 2012), led to significant increases in water storage in the study area. From September 2009 to June 2010, positive trends were observed: ΔTWS increased at 2.3 cm/month, ΔSMS at 1.9 cm/month, and ΔGWS exhibited an increasing trend of 0.9 cm/month; all trends are statistically significant (p < 0.05). These trends confirm that La Niña events promote recharge

effectively counteracting the deficits induced by preceding El Niño events.

Although GRACE-derived ΔGWS provides a valuable large-scale perspective on groundwater storage changes, piezometric data from monitoring wells offer localized insights into aquifer responses to ENSO variability. Limited groundwater level changes (ΔGWL) data from the GM and TS sites (Fig. 8) confirm that the 2015-2016 El Niño event was associated with significant groundwater level declines, suggesting reduced recharge rates during this period. Following this

water storage depletion phase, a notable recharge event occurred in 2017, coinciding with the transition to La Niña conditions which facilitated moisture recovery in soils and aquifers. Similarly, the 2020-2022 La Niña events (Shi et al., 2023), typically associated with wetter conditions in Southeast Asia (Feng and Wang, 2018), coincide with increased ΔTWS, ΔSMS, and ΔGWS. Groundwater levels at monitoring sites MM and TS also exhibited positive trends, reinforcing the role of La Niña in replenishing regional water resources. These underscore the dynamic relationship between ENSO phases and terrestrial water

storage components in the study area. El Niño events lead to significant reductions in ΔTWS and ΔSMS, resulting in drought conditions and lower groundwater recharge rates whereas La Niña events promote water storage recovery, characterized by increases in precipitation, soil moisture retention, and groundwater recharge.



### 4.3 Uncertainty in GRACE-based ΔTWS and ΔGWS estimates

The GRACE JPL ΔTWS dataset employed in this study provides conservative uncertainty estimates (Wahr et al., 2006) for
the effective 3° grid (mascon). The mean uncertainty for the mascon encompassing the study area is ~2 cm, with a standard
deviation of ~1 cm (Fig. S16). However, uncertainty estimates for the finer-resolution 0.5° GRACE JPL grid are unavailable
and are expected to be higher due to greater variability at smaller spatial scales (Landerer and Swenson, 2012). The absence
of refined uncertainty quantifications at sub-mascon scales poses a limitation for ΔGWS assessments in localized settings as
hydrological variability increases with decreasing spatial scale, making ΔGWS more sensitive to data noise and model
parameterization errors.

As noted earlier, two of the eight 0.5° GRACE JPL grids within the study area (Fig. S11) exhibit negative gain factors (-
0.07), resulting in systematically lower ΔTWS magnitudes over time (Fig. S10a). This dampening effect is not observed in the
GRACE GSFC and CSR datasets and is likely caused by oceanic signal interference with terrestrial water storage (Landerer
and Swenson, 2012). In coastal areas such as the Mahakam Delta, the proximity to ocean masses can introduce filtering effects
that suppress ΔTWS variability, reducing observed fluctuations. As a result, in areas dominated by negative gain factor grids,
ΔSMS arithmetically becomes the primary driver of ΔTWS variability, which can lead to overestimation or underestimation
of ΔGWS as the residual parameter.

Computational uncertainty in GRACE-derived ΔGWS arises due to its estimation as a residual component derived from
the difference between GRACE-based ΔTWS and simulated water storage components (Eq. 1). Since ΔGWS is not directly
measured by satellites but inferred through hydrological balance calculations, its accuracy depends on the reliability of ΔTWS
estimates and the other simulated water components in the study area, primarily ΔSMS and ΔSWS.

As discussed in Section 3.2, ΔSMS exerts the most substantial influence on ΔTWS in this study, whereas ΔSWS
contributes minimally, and ΔCW contributions are negligible (Table S4). Because ΔGWS is computed as a residual, any
discrepancy between ΔTWS and ΔSMS directly affects ΔGWS estimations (Shamsudduha and Taylor, 2020). A key
observation in Fig. 4 highlights that when ΔSMS values are lower than ΔTWS, ΔGWS estimates become anomalously high.

Another major source of uncertainty in GRACE-derived ΔGWS calculations is the representation of surface water storage
in GLDAS datasets. Notably, ΔSWS in GLDAS is not a direct measure of surface water storage but rather a simulation of
surface runoff (Beaudoing and Rodell, 2020; Rodell et al., 2004; Li et al., 2019). This distinction is critical because surface
runoff estimates do not account for standing water bodies (rivers, lakes, reservoirs, and wetlands), potentially leading to an
underestimation of total ΔSWS contributions to ΔTWS (Scanlon et al., 2019; Getirana et al., 2017; Shamsudduha et al., 2012;
Proulx et al., 2013). This limitation is particularly relevant in tropical regions such as the Amazon Basin where surface water
storage contributes up to ~27% of total ΔTWS variations (Getirana et al., 2017).

In the present study, mean ΔSWS estimates from the Catchment LSM are significantly higher (-10.7 to 20.3 cm) than
those from the WGHM and Noah LSM, which range from -3.7 to 4.1 cm (Fig. S1). This discrepancy reinforces that using
ΔSWS from the Catchment LSM could introduce substantial uncertainty into ΔGWS calculations. In contrast, ΔSWS from



WGHM and Noah LSM appears more reasonable and is further supported by global surface water extent data (Pekel et al., 2016) which show considerable temporal variability in the study area (Fig. S1).

Across the 36 realizations employed in this study, ~30% of GRACE-derived ΔGWS estimates per realization are implausible (Table S5) during the wet season when ΔTWS, ΔSMS, and ΔSWS are all positive and ΔTWS is lower than the

combined ΔSMS and ΔSWS and during the dry season when all components are negative, the absolute value of ΔTWS is smaller than the sum of the absolute values of ΔSMS and ΔSWS. This study emphasizes the importance of removing implausible water storage components from subsequent analyses as their inclusion could introduce significant bias and errors into hydrological interpretations and climate-groundwater assessments.

## 4.4 The role of extreme rainfall in groundwater recharge

This study finds moderate correlations between mean ensemble ΔTWS, ΔSMS, and monthly precipitation averaged from Balikpapan and Samarinda stations. The correlation coefficients are 0.5 for both ΔTWS and precipitation as well as ΔSMS and precipitation. Introducing a 1-month lag does not significantly alter these values. In contrast, longer lags lead to a substantial decrease in correlation, dropping to 0.2 for both ΔTWS and ΔSMS at a 3-month lag. These correlations suggest a relatively strong coupling with minimal lag effects between precipitation inputs and terrestrial water storage in the study area,

particularly in near-surface soil moisture storage dynamics. This may indicate rapid infiltration of rainfall into the upper soil layers, likely due to limited water retention capacity in the vadose zone or the presence of highly permeable soils.

The computed mean ensemble ΔGWS exhibits a weak correlation with precipitation (r = -0.33). Introducing 1- to 3-month lags does not alter the correlation significantly, whereas longer lags lead to a further decline, with r = 0.13 at a 4-month lag. Despite the weak correlation between ΔGWS and precipitation, piezometric data reveal the important role of extreme

rainfall events in groundwater recharge in the study area. The comparison between hourly piezometric data and daily precipitation (Fig. 9) confirms relatively rapid groundwater replenishment, driven by extreme rainfall events.

The contribution of high-intensity precipitation to groundwater recharge has been widely documented in recent studies (Shamsudduha and Taylor, 2020; Taylor et al., 2013a; Cuthbert et al., 2019b; Jasechko and Taylor, 2015; Goni et al., 2021; Seddon et al., 2021; Jasechko, 2019). Observations from January 2024 to February 2025 indicate that extreme daily

precipitation events exceeding the 90th percentile (>35 mm) are consistently followed by groundwater level rises (Fig. 9), reinforcing the notion that episodic high-intensity rainfall contributes to groundwater recharge in the study area. This evidence underscores the critical role of extreme rainfall events in groundwater recharge, suggesting that increased rainfall due to climate change (Taylor et al., 2013b) may enhance future groundwater recharge in the study area.



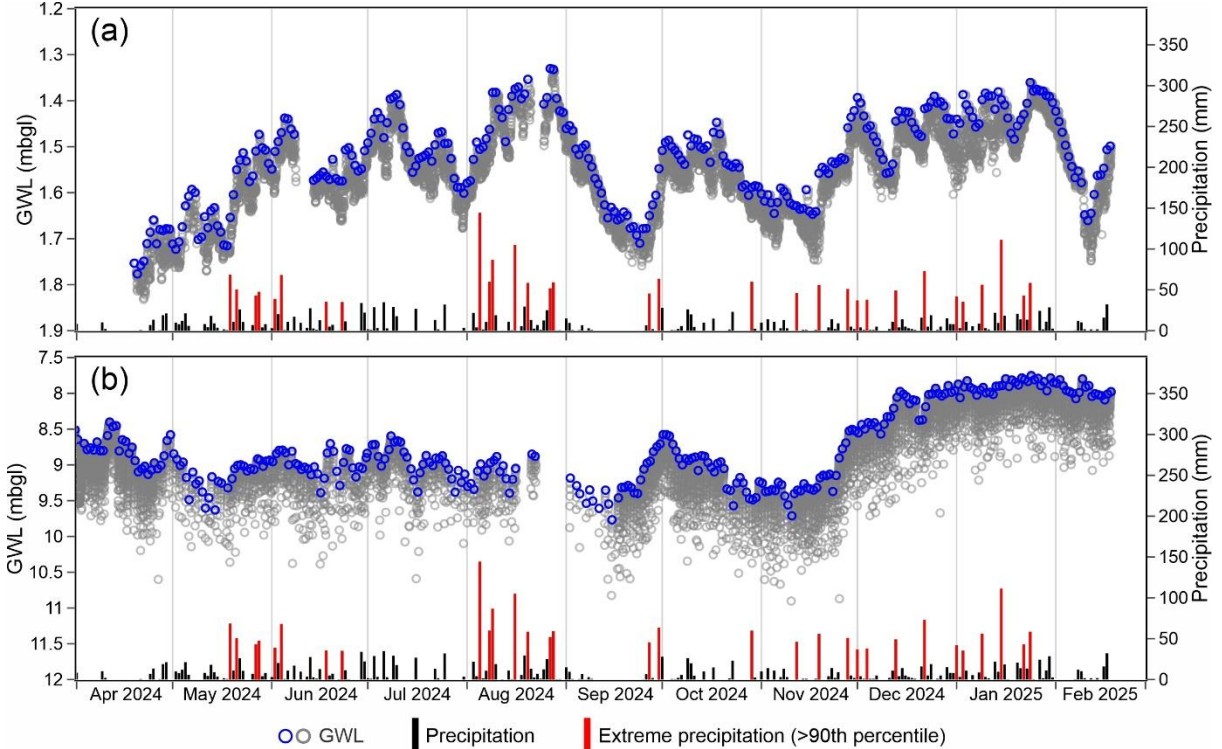

**Figure 9: Comparison of hourly groundwater level (GWL) with daily precipitation from Balikpapan station at two monitoring sites: (a) MG site (screened interval: 30-33 m) and (b) EBD site (screened interval: 73-98 m). Blue circles represent the minimum GWL recorded per day. In (b), gray circles denote recovery data, whereas blue circles indicate the minimum GWL per day.**

### 4.5 Challenges in groundwater monitoring and the potential of pumping well data

Establishing an extensive groundwater monitoring network in Indonesia including the study area is required to advance the renewability of groundwater withdrawals, particularly as many global aquifers experience rapid depletion (Jasechko et al., 2024; Taylor, 2014). An extensive monitoring system would facilitate early detection of depletion trends, inform water resource planning and support evidence-based policy decisions to prevent overabstraction. This is especially important in regions where geological heterogeneity and anthropogenic influences such as groundwater abstraction lead to significant

groundwater level variations that are not well captured by large-scale GRACE ΔTWS and associated estimates of ΔGWS. Although the integration of GRACE ΔTWS with the estimation of other individual water storage components such as soil moisture (ΔSMS) from GLDAS (Rodell et al., 2004) and surface water storage (ΔSWS) from WGHM (Müller Schmied et al., 2024), provides valuable regional-scale ΔGWS estimates, their reliability depends on the accuracy of filtered GRACE data and simulated water storage components. Any anomalous discrepancy between ΔTWS and the simulated water storage

components directly affects ΔGWS calculations and can introduce substantial arithmetic errors. These limitations constrain the applicability of GRACE-derived ΔGWS for detecting localized groundwater fluctuations.



Limited piezometric observations in Balikpapan City, south of Indonesia's new capital (Nusantara), highlight the spatial and temporal variability in groundwater levels (Fig. 8, 9, and S6). Lithological logs from monitoring (MG) and pumping (EBD) sites (Fig. S17) reveal alternating sequences of sand- and clay-dominated layers. Both sites exhibit near-surface clay layers (0-70 m), transitioning into a sand-dominant sequence (70-105 m). At MG, deeper lithology (105-130 m) reverts to clay-dominant deposits. The shallowest piezometer (screened at 30-33 m) at the MG site exhibits minimal fluctuations (1.3-1.8 mbgl) between April 2024 and February 2025, correlating with precipitation patterns particularly extreme rainfall events that contribute to groundwater recharge (Fig. 9a). In contrast, deeper piezometers at the same site show distinct responses. At 87-90 m depth, groundwater levels remained stable until May 2024, when a sudden ~10 m rise occurred, stabilizing thereafter (Fig. S18). This pattern suggests a significant reduction of abstraction, allowing for groundwater recovery. Conversely, groundwater levels at 132-135 m depth exhibit a continuous decline of -1.5 cm/month ($p < 0.05$), indicating persistent abstraction. Additionally, groundwater levels at this site show evidence of poroelastic aquifer responses (Burgess et al., 2017), likely influenced by tidal loading and unloading as the piezometer is located ~400 m from the coastline. These observations underscore the necessity of high-resolution, depth-specific monitoring to differentiate between natural and anthropogenic impacts.

The limited availability of groundwater level data in the study area restricts basin-wide assessments of spatial and temporal groundwater storage changes, thereby constraining efforts to inform renewable groundwater use. This limitation highlights the urgent need for an expanded monitoring infrastructure. A comparison with other countries illustrates the scale of monitoring required: Austria maintains ~3,535 groundwater observation points, Bangladesh has over 8,000 monitoring wells, and the United States operates more than 14,000 stations (IGRAC, 2020). However, financial and logistical constraints make large-scale groundwater monitoring challenging. An alternative approach explored in this study involves leveraging existing groundwater pumping wells for monitoring. This strategy can provide continuous groundwater level observations in areas lacking dedicated monitoring wells and significantly reduce costs (Chilton and Foster, 2024).

To evaluate this approach, a water logger was installed in an active pumping well in Balikpapan City, where groundwater abstraction is substantial (Fig. S15). Similar techniques have been proposed by Abi et al. (2024), who developed methods for filtering out data collected during active pumping and imputing missing values to maintain continuous groundwater level records. The effectiveness of this method was assessed by comparing groundwater level fluctuations at the EBD pumping well (screened 73-98 m) with those at the MG dedicated monitoring site. Groundwater levels at EBD exhibited clear responses to precipitation, showing recharge following extreme rainfall events, closely mirroring trends at MG. By filtering out data recorded during pumping periods and retaining only the minimum daily groundwater levels from recovery phases, an approximated groundwater level time series was extracted from the pumping well (Fig. 9b). These suggest that with appropriate data filtering techniques, pumping wells can serve as a viable and cost-effective alternative to traditional monitoring wells, offering a scalable solution for groundwater monitoring in Indonesia.



## 5 Conclusions

In a coastal (island) environment in the humid tropics of Indonesia, we examine the applicability of GRACE satellite data to estimate groundwater storage changes (ΔGWS) in a data-limited location that is below the native spatial resolution of GRACE (≥90,000 km$^2$). A comparative evaluation of a range of GRACE products with different spatial scales shows strong correlations (r = 0.85 to 1) and consistency, with mean ensemble terrestrial water storage anomalies (ΔTWS) ranging from -15.9 to 12.9 cm. Among the simulated water storage components used to compute GRACE ΔGWS that derive from global-scale models, soil moisture storage changes (ΔSMS) contribute the most to ΔTWS, followed by surface water storage changes (ΔSWS); canopy water anomalies (ΔCW) are negligible. We note that of the 36 realizations derived from the employed range of GRACE solutions and global-scale models, ~30% of GRACE-derived ΔGWS estimates in each realization are physically implausible, exhibiting positive values during dry periods and negative values during wet periods. Mean ensemble values of GRACE ΔGWS range from -5.7 to 8.7 cm, with a statistically significant annual linear trend of 0.2 cm/year (p-value < 0.05). GRACE ΔGWS time series correspond moderately (r = ~0.4) with groundwater level changes (ΔGWL) from piezometry; observed deviations from GRACE ΔGWS are associated with groundwater withdrawals for domestic use.

Statistical analyses reveal moderate correlations (r = -0.4 to -0.6) between ΔTWS, ΔSMS, and ENSO indices. In contrast, ΔGWS shows weaker correlations with climate indices, with the strongest observed between ΔGWS and MEI (r = -0.35). No significant lag effects were observed between water storage components (ΔTWS, ΔSMS, ΔGWS) or precipitation and ENSO indices, as correlations weakened with increasing lag time. Piezometric data confirm responses to ENSO variability, particularly during the 2015-2016 El Niño and the 2020-2022 La Niña events, with El Niño associated with water storage deficits and La Niña promoting recharge. Comparisons of hourly piezometric data with daily precipitation records reveal episodic, high-intensity rainfall events play a significant role in generating groundwater recharge.

The absence of an extensive groundwater monitoring network in Indonesia including in the data-scarce Lower Kutai Basin (LKB) where Indonesia's new capital (Nusantara) is under development, limits the ability to track long-term groundwater trends and inform effective groundwater management. To address this, we explore the use of observations from pumping wells for groundwater monitoring. Through the application of data-filtering techniques to isolate recovery periods from active pumping intervals, approximated groundwater level time series are derived; this approach offers a cost-effective, scalable solution for enhancing groundwater monitoring in Indonesia, particularly in areas where logistical and financial constraints hinder the deployment of dedicated monitoring wells.

## Data availability

Satellite data, global-scale models, and climate indices used in this study are available from the sources provided in the text.



**Author contribution**

A, RGT, and MS designed the study. A conducted the analyses and visualization under the supervision of RGT and MS. AMR contributed to data provision and analysis. A drafted the manuscript with input from all co-authors.

**Competing interests**

The authors declare that they have no conflict of interest.

**Acknowledgements**

The authors thank the Indonesia Endowment Fund for Education (LPDP) for funding this study. RGT acknowledges support from a Fellowship (ref. FL-001275) provided by the Canadian Institute for Advanced Research (CIFAR) under the Earth 4D
program.

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
