# Peer review of "Groundwater storage dynamics and climate variability in the Lower Kutai Basin of Indonesia: reconciling GRACE $\Delta$ GWS to piezometry"

_EGUsphere, 2025_

## Author Comment (AC1)

**Responses to reviewer's comments**

We appreciate the constructive comments of Jürgen Kusche. In the revised manuscript, we have incorporated changes (highlighted in blue) to address all comments presented below in *italics*. Our responses to each comment are provided in **bold**. Please note that line numbers are provisional and may change in the final revised manuscript after receiving comments from all reviewers.

**General**

*Arifin et al look at three different ways of measuring groundwater storage dynamics in the Lower Kutai Basin (LKB) of Indonesia: the GRACE budget residual approach, data from the existing (few) local piezometric sensors, and the possibility to use the sensors from pumping wells. The study is motivated by the fact that Indonesia's new capital Nusantera is under development in the LKB which will lead to increased pressure on water resources.*

*The main message of the paper seems to me that currently neither of the three approaches can provide a reliable assessment of groundwater resource variability, and the government should think about rolling out a comprehensive measuring network. In my understanding the region is simply too small to be resolved well in GRACE (as the authors know very well), and there are too few in-situ sensors. The authors suggest that about 30% of their GRACE-derived dGWS ensemble timeseries are not usable since they provide unphysical results, and they discuss many timeseries with correlations about 0.3 as „weakly correlated" – I think given that timeseries here are not very long on climate timescales we can say that such low correlation means nearly uncorrelated.*

*On balance, my judgement is that the topic is very relevant and there seems a pressing need to improve the monitoring system, however the quantitative evidence that is presented is a bit weak and the logic is not straightforward. It is clear that the region is not well suited for a thorough assessment of the GRACE data, or the GRACE residual approach for groundwater, since the basin is too small, the GRACE data are affected by ocean signals, and there are too few in-situ sensors. I am missing a discussion of related papers that discuss bigger inland regions of similar hydrogeophysics but better monitoring network with a similar approach. If I am right about the authors' intention, I am missing a much more extended discussion of how a monitoring network could look like, how many piezometers or observation wells would be needed, how could they be distributed to connect to the GRACE data. Without this I feel the paper misses a bit its mark.*

We thank the reviewer for the thoughtful and constructive review. We agree that the study area (Lower Kutai Basin, LKB) lies below the spatial resolution of GRACE and that limited piezometric coverage constrains robust quantification of groundwater storage dynamics. Indeed, our aim is to evaluate the applicability and limits of GRACE in small, coastal, and inadequately monitored regions. Demonstrating this limitation, though expected, is an important result in itself, given the need to assess and monitoring groundwater storage underlying Indonesia's new capital, Nusantara, in the LKB.

We have added a discussion of the Bengal Basin, which shares a similar coastal deltaic setting with the LKB but benefits from much denser monitoring networks (lines 479-481):
*"In the Bengal Basin (~138,000 km$^2$), which shares a similar coastal deltaic setting as the LKB, ΔGWS estimates derived from a dense network of monitoring wells exhibit a strong correlation with GRACE-based ΔGWS (Shamsudduha et al., 2012). In other regions however, GRACE-based ΔGWS align less well with observed values."*

We have also expanded our discussion of what a future monitoring network in the LKB could look like. Specifically, we suggest the number of wells (~70) and their spatial density (~1 per 50 km$^2$ in urban areas and ~1 per 500 km$^2$ in rural or less populated areas) (lines 682-686):
*"For the study area, we suggest ~70 monitoring wells, one well per ~50 km$^2$ in urban areas such as Balikpapan and Samarinda and one per ~500 km$^2$ in rural or less populated regions. The final network will also need to consider proximity to major groundwater withdrawals and discharge zones, representation of key hydrostratigraphic units, and water-quality risk areas; such a network could support validation of GRACE-derived ΔGWS estimates and inform groundwater management in Nusantara."*

**Responses to specific points are addressed below:**

*1. Title: I don't think the authors really succeed in „reconciling GRACE to piezometry". This should be reflected in the title.*

**R1: We thank the reviewer for this helpful suggestion. We agree that our study highlights limitations, rather than successful reconciliation of GRACE-derived**

**groundwater storage changes with sparse piezometric observations. Accordingly, we have revised the title to better reflect the scope and findings of the study:** "*Groundwater storage dynamics and climate variability in the Lower Kutai Basin of Indonesia: challenges to the reconciliation of GRACE ΔGWS and piezometry*"

*2. The authors discuss the challenges of the GRACE residual approach, e.g. we need to know the soil moisture contribution. But we also need to know the contribution by surface water storage variability, water levels in wetlands and rivers, lakes and artificial reservoirs. The authors are aware of this. In my understanding, wetlands cover a significant share of land in Indonesia including the LKB. We need a quantitative discussion of the error that could be introduced here. WaterGAP may not be very good at this – WaterGAP does not simulate level changes in reservoirs, for example. And I suggest that the authors look into ways of quantifying this contribution, e.g. from remote sensing and/or radar altimetry. Similar for hydrology, in my understanding in Indonesia peatlands are abundant and peatland hydrology may not be well represented in GLDAS or WaterGAP. What is the anticipated error? Or isn't this the case in the LKB?*

**R2: We thank the reviewer for highlighting these important points. We agree that contributions from rivers, wetlands, lakes, and reservoirs introduce significant uncertainty into GRACE residual groundwater storage (ΔGWS) estimates. To address this, we have substantially revised discussion of surface water storage changes (ΔSWS) on lines 163-166, 330-345, and 609-616.**

**Lines 163-166 (Data – 2.3 GLDAS and WGHM data):**
**"***In addition, we employ the global lakes bathymetry (GLOBathy) dataset from Khazaei et al. (2022) and global surface water extent data from Pekel et al. (2016) to estimate lake water storage changes (ΔLS) in the study area. Schwatke et al. (2015) provide global river water level data from satellite altimetry yet only one station is available in the LKB with limited temporal coverage.***"**

**Lines 330-345 (Results – 3.2 Simulated water storage components from GLDAS and WGHM):**
**"***ΔSWS was derived from WGHM and Noah LSM, ranging between -1 to 3.7 cm for Noah and -5.3 to 5.5 cm for WGHM. The broader range in WGHM reflects stronger variability in simulated surface water processes. Using bathymetry data from 15 lakes in GLOBathy dataset (Khazaei et al., 2022), representing the major surface water bodies in the study area (Fig. S15) together with surface water extent changes from Pekel et al.***

*(2016), we estimated lake storage changes (ΔLS). ΔLS values are generally smaller than ΔSWS from both Noah and WGHM, ranging from -1.6 to 0.8 cm (Fig. S16). ΔLS from GLOBathy shows moderate agreement with WGHM-estimated ΔLS (r = 0.52; RMSE = 0.1 cm) and ΔSWS from Noah (r = 0.53; RMSE = 0.1 cm) but a weaker agreement with ΔSWS from WGHM (r = 0.12; RMSE = 0.4 cm).*

*River storage changes could not be quantified because of the sparse river monitoring network (Schwatke et al., 2015). The global surface water dataset (Pekel et al., 2016) indicates that river areas in the LKB varied between 30 and 125 $km^2$ (mean ~110 $km^2$) between 2003 and 2021 (Fig. S16), whereas lake areas varied between 10 and 370 $km^2$ (mean ~260 $km^2$). These differences suggest that river storage changes were likely smaller than ΔLS, though this remains unverified due to lack of volume estimates. In addition, wetlands and peatlands surrounding the lakes in the northwest of the study area (Patria et al., 2025) may contribute to underestimation of ΔSWS. On average, lakes and rivers account for ~77% of the maximum surface water extent (Fig. S16), with wetlands and peatlands comprising the remaining ~23%. WGHM simulates negligible wetland storage variability (on the order of $10^{-2}$ cm), indicating that current models may not adequately capture wetland and peatland dynamics.*"

**Lines 609-616 (Discussion – 4.3 Uncertainty in GRACE-based ΔTWS and ΔGWS estimates):**

"*In contrast, ΔSWS from WGHM and Noah LSM appear more representative and are further supported by lake storage changes (ΔLS) derived from bathymetry of 15 lakes in the GLOBathy dataset (Khazaei et al., 2022) and surface water extent changes from Pekel et al. (2016). ΔLS values show relatively good agreement with ΔSWS from Noah LSM (RMSE = 0.1 cm) and WGHM (RMSE = 0.4 cm). Although river storage changes cannot be quantified directly due to sparse monitoring (Schwatke et al., 2015), the global surface water dataset (Pekel et al., 2016) suggests that river storage changes may be smaller than ΔLS and thus contribute less to ΔSWS than lakes. In addition, the lack of robust estimates for changes in wetlands and peatlands storage may lead to an underestimation of ΔSWS as these systems account for ~23% of surface water extent changes.*"

*3. Similar, the Makassar and Sulawesi Straits are part of an ocean region that experiences above-average variability and sea level rise. Ocean signals are reduced in the standard GRACE data products but we know that the MPI reference ocean model which is forced by the atmosphere only does not capture all mass signals, and in coastal areas this may introduce a significant error irrespectively what mascon products were used. Put in simple words, the*

*GRACE data may include real ocean mass change at least on the seasonal timescale here that could be misinterpreted as a groundwater signal. This is an error source that is not relevant for the average hydrological inland basin, but for Indonesia it may be very well. I'm missing a quantitative discussion here.*

**R3: We thank the reviewer for raising this critical comment. We agree that in the Indonesian maritime setting, particularly along the Makassar and Sulawesi Straits, strong ocean mass variability could introduce a distinct source of error into GRACE land grids. Standard GRACE products (mascons and spherical harmonics) attempt to reduce ocean leakage using ocean models, however, leakage may still persist. We have expanded our discussion of potential ocean leakage in lines 300-310:**

**"*We further assessed leakage by pairing each Borneo land grid with the nearest ocean grid and computing correlations before and after detrending and deseasonalizing using STL (Fig. S12). Residual land-ocean correlations remain non-negligible and they differ by product. Within 100-150 km of the coast, CSR shows the highest median residual correlation (0.68; $r^2 = 0.46$), whereas JPL (0.22; $r^2 = 0.05$) and GSFC (0.17; $r^2 = 0.03$) are much weaker. GFZ (0.43; $r^2 = 0.18$) and COST-G (0.36; $r^2 = 0.13$) fall in between. Inland (150-250 km), CSR weakens (0.42; $r^2 = 0.17$), whereas GFZ (0.47; $r^2 = 0.22$) and COST-G (0.42; $r^2 = 0.17$) retain relatively moderate correlations that may reflect leakage and filtering artifacts. These results demonstrate that leakage effects are strongly product dependent, with CSR most affected near the coast and spherical harmonic products retaining inland correlations. These correlations represent, however, only an upper boundary since shared land-ocean variance may also reflect co-varying climate signals and ocean loading due to the non-unique nature of mass inversion (Heki and Jin, 2023; Ndehedehe and Ferreira, 2020; Chen et al., 2022; Chao, 2005). Identifying the relative contributions of different sources requires independent constraints from numerical modeling or alternative observational methods (Chen et al., 2022).*"**

*4. The authors should also try to estimate the error in the piezometric analysis introduced by not knowing the local yield factor. They could take a range of possible yield factors from the hydrogeophysics maps or from publications and do a best/worst-case assessment.*

**R4: We agree that uncertainty in storage coefficients introduces substantial error into the conversion of groundwater level anomalies (ΔGWL) to storage changes (ΔGWS). In the revision, we have expanded our analysis to consider the implications of employing a range of storage coefficients (lines 492-502):**

*"In the study area, groundwater abstraction is concentrated within ~20 km of coast, particularly south of Balikpapan City (Fig. S23). We compared ensemble mean plausible GRACE-derived ΔGWS with groundwater level changes (ΔGWL) as the storage coefficient (S or Sy) is not well constrained. Lithological logs reveal heterogeneous interbedded sand and clay units (Fig. S24; Arifin et al. (2024)). Piezometer depths range from 21 to 135 m; shallow screens up to ~40 m may represent unconfined aquifers, whereas deeper screens may tap confined aquifers. In similar deltaic settings such as the Mekong and Indo-Gangetic deltas, storage coefficients mostly vary from ~0.08 to 0.25 (mean ~0.15) for unconfined aquifers and from ~$10^{-5}$ to $8 \times 10^{-4}$ (mean ~$5 \times 10^{-4}$) for confined aquifers (BGS and DPHE, 2001; Bonsor et al., 2017; Van et al., 2023; Pechstein et al., 2018). This uncertainty translates into a range of possible ΔGWS values (Fig. 8). On average, confined aquifers yield ~0.5 mm of storage loss per metre of decline (S = $5 \times 10^{-4}$), whereas unconfined aquifers yield ~15 cm (Sy = 0.15). Although correlations are unaffected by this uncertainty, the amplitude of GRACE-derived ΔGWS remains highly uncertain without reliable storage coefficients."*

*5. The comparison of the GRACE product error with three mascon solutions appears not very robust. I would very much recommend that the authors consider at least one product based on spherical harmonics. It is an unproven claim that mascon solutions are better or more suitable to coastal regions. They are easy to apply but this is not the same as being more appropriate or having less errors.*

**R5: We appreciate this comment and agree that our evaluation should not be limited to mascon products nor suggest that mascons are inherently "better" for coastal regions. In the revision, we have incorporated two spherical harmonic (SH) solutions, GFZ and COST-G.**

*6. I'm missing a map of the aquifer systems in the LKB, in particular are these aquifers extending under the sea? The dashed line in Fig. 1 suggests this. Would coastal groundwater withdrawal then cause storage changes in the marine part of the aquifer? Would that be expected to become visible in GRACE as well? Isn't this suppressed by the mascon approach? These are just ideas, I am not an expert in this regions. Again, most GRACE groundwater studies need not worry about this, but the region here is particularly challenging.*

**R6: A short description has been provided in lines 115-118; a hydrogeological map and cross-section in Figs. S2-S3 (Supplementary) illustrate aquifer distributions in the LKB and their potential continuation offshore.**

**Lines 115-118:**

*"Arifin et al. (2024) provide surface geological and hydrogeological maps of the coastal LKB (Fig. S2). The regional hydrostratigraphy is primarily composed of Miocene to Quaternary deltaic deposits which are extensively distributed across the coastal LKB (KESDM, 2022; Moss and Chambers, 1999). These deposits mainly consist of interbedded sand and clay layers, forming a complex aquifer system that may extend offshore (Fig. S3) toward the Makassar Strait (Arifin et al., 2025)."*

*7. There are GRACE-assimilating hydrology model runs from NASA/Goddard and from the University of Bonn, Germany, and these provide the groundwater storage change at resolution between 30 and 50 km. Why don't the authors look into these data sets or add them to their ensembles?*

**R7: We thank the reviewer for highlighting GRACE-assimilating models. In the revision, we have included ΔGWS estimates from GLWS 2.0 assimilation product which integrates GRACE ITSG into WaterGAP, as well as the GLDAS-2.2 daily product which assimilates GRACE CSR into Catchment LSM.**

*8. The authors mention several times that poor GRACE data or poor corrections in the residual approach lead to arithmetic problems. This is true, but an arithmetic problem is just a symptom that the data are poor. In other words, even if no arithmetic problem occurs this may be just by chance, and we should not trust the data. I think this needs to be made clear in a scientific paper.*

**R8: We thank the reviewer for this important clarification. We agree that the occurrence of arithmetic problems may indicate poor or inconsistent data inputs, and the absence of such problems does not guarantee that the underlying GRACE or model-simulated components are reliable. We have made revisions in lines 617-622:**

*"Across the 54 realizations tested in this study, ~42% of GRACE-derived ΔGWS estimates per realization are implausible, typically appearing as negative values during wet periods when all components are positive, or positive values during dry periods when all components are negative. Including implausible values reduces the correlation between ΔGWS and ΔTWS from 0.86 to 0.12 (Fig. 6a, b). The implausible values may not be merely computational anomalies but rather symptoms of poor or inconsistent input data, particularly where modeled ΔSMS and ΔSWS diverge from GRACE-observed ΔTWS (Scanlon et al., 2018)."*

*9. Potential instrumental problems of the piezometric sensors should be discussed. I liked the part on the correlation between these data and the rainfall data, as it tells about the sensitivity. But this instigates trust for the short timescales only. What about the seasonal and interannual timescales, are there biases to be expected?*

**R9: We agree that although short-term sensitivity of piezometers to rainfall inputs provides confidence in their functionality, instrumental problems may introduce biases on seasonal and interannual timescales. We have expanded our discussion of potential instrumental limitations in lines 699-702:**

***"However, reliability requires periodic manual validation of piezometric sensors, such as monthly checks during the first few months of deployment and semi-annual checks thereafter, in order to mitigate instrumental issues such as sensor drift and calibration errors that can bias long-term records at seasonal or interannual timescales."***

*10. Overall, the error budgeting needs more detail and quantification. This is, as I said earlier, partly a consequence of the fact that the LKB region is a particularly challenging one for GRACE. That also means if the authors succeed to make their case, this could be a breakthrough in the application of GRACE data, so it is really worth to dig deeper. At the moment, results appear somewhat inconclusive and the message is not too clear. I suggest that the study logic – what is the underlying hypothesis, what exactly do we expect from GRACE at such small scales, why looking at the piezo-rainfall correlation, why looking at ENSO – is explained right at the start.*

**R10: We thank the reviewer for this constructive feedback. We agree that clearer framing of the study logic and more transparent error budgeting are essential. We have re-reviewed the prose of each section to more clearly articulate the merits of this study and its findings. In addition, we have rewritten the last paragraph of the introduction (lines 88-95) to provide clearer context for the study.**

**Lines 88-95:**
***"This study examines whether GRACE can provide physically meaningful signals of groundwater storage variability in the small, coastal, and data-scarce Lower Kutai Basin (LKB), recognizing that its spatial scale lies below GRACE's effective resolution and is highly susceptible to ocean leakage. Specifically, we (1) compare ΔTWS across multiple GRACE products, (2) evaluate the plausibility of GRACE-derived ΔGWS against limited piezometric data, and (3) assess whether large-scale climate drivers,***

*particularly ENSO, are detectable. The aim is to evaluate the limits of GRACE in this challenging setting and to demonstrate that without a substantially expanded piezometric network, neither GRACE nor in situ observations alone can provide robust groundwater storage assessments to support the development of climate-resilient groundwater management strategies, particularly in rapidly urbanizing regions such as Nusantara where water security is a growing concern."*

**References**

Arifin, Shamsudduha, M., Ramdhan, A. M., Reksalegora, S. W., and Taylor, R. G.: Characterizing deep groundwater using evidence from oil and gas exploration wells in the Lower Kutai Basin of Indonesia, Hydrogeology Journal, 32, 1125-1144, 10.1007/s10040-024-02776-0, 2024.

Arifin, Taylor, R. G., Shamsudduha, M., Ramdhan, A. M., Iskandar, I., Setiawan, T., Iman, M. I., and Noor, R. A.: Hydrochemistry of a coastal sedimentary basin: evidence from the Lower Kutai Basin, Indonesia, Applied Geochemistry, 190, 106496, 10.1016/j.apgeochem.2025.106496, 2025.

BGS and DPHE: Arsenic contamination of groundwater in Bangladesh, in: British Geological Survey Technical Report WC/00/19, edited by: Kinniburgh, D. G., and Smedley, P. L., British Geological Survey, Keyworth, 2001.

Bonsor, H. C., MacDonald, A. M., Ahmed, K. M., Burgess, W. G., Basharat, M., Calow, R. C., Dixit, A., Foster, S. S. D., Gopal, K., Lapworth, D. J., Moench, M., Mukherjee, A., Rao, M. S., Shamsudduha, M., Smith, L., Taylor, R. G., Tucker, J., van Steenbergen, F., Yadav, S. K., and Zahid, A.: Hydrogeological typologies of the Indo-Gangetic basin alluvial aquifer, South Asia, Hydrogeology Journal, 25, 1377-1406, 10.1007/s10040-017-1550-z, 2017.

Chao, B. F.: On inversion for mass distribution from global (time-variable) gravity field, Journal of Geodynamics, 39, 223-230, 10.1016/j.jog.2004.11.001, 2005.

Chen, J., Cazenave, A., Dahle, C., Llovel, W., Panet, I., Pfeffer, J., and Moreira, L.: Applications and Challenges of GRACE and GRACE Follow-On Satellite Gravimetry, Surveys in Geophysics, 43, 305-345, 10.1007/s10712-021-09685-x, 2022.

Heki, K. and Jin, S.: Geodetic study on earth surface loading with GNSS and GRACE, Satellite Navigation, 4, 24, 10.1186/s43020-023-00113-6, 2023.

KESDM: National geologic map and aquifer productivity [dataset], 2022.

Khazaei, B., Read, L. K., Casali, M., Sampson, K. M., and Yates, D. N.: GLOBathy, the global lakes bathymetry dataset, Scientific Data, 9, 36, 10.1038/s41597-022-01132-9, 2022.

Moss, S. J. and Chambers, J. L. C.: Tertiary facies architecture in the Kutai Basin, Kalimantan, Indonesia, Journal of Asian Earth Sciences, 17, 157-181, 10.1016/S0743-9547(98)00035-X, 1999.

Ndehedehe, C. E. and Ferreira, V. G.: Identifying the footprints of global climate modes in time-variable gravity hydrological signals, Climatic Change, 159, 481-502, 10.1007/s10584-019-02588-2, 2020.

Patria, A. A., Obrochta, S. P., and Anggara, F.: Tracing highly oxidized events and its response to peat dynamic from the northwest Kapuas coastal wetlands, Indonesia, International Journal of Coal Geology, 303, 104751, 10.1016/j.coal.2025.104751, 2025.

Pechstein, A., Hanh, H. T., Orilski, J., Nam, L. H., and Manh, L. V.: Detailed Investigations on the hydrogeological situation in Ca Mau Province, Mekong Delta, Vietnam, 2018.

Pekel, J.-F., Cottam, A., Gorelick, N., and Belward, A. S.: High-resolution mapping of global surface water and its long-term changes, Nature, 540, 418-422, 10.1038/nature20584, 2016.

Scanlon, B. R., Zhang, Z., Save, H., Sun, A. Y., Müller Schmied, H., van Beek, L. P. H., Wiese, D. N., Wada, Y., Long, D., Reedy, R. C., Longuevergne, L., Döll, P., and Bierkens, M. F. P.: Global models underestimate large decadal declining and rising water storage trends relative to GRACE satellite data, Proceedings of the National Academy of Sciences, 115, E1080-E1089, 10.1073/pnas.1704665115, 2018.

Schwatke, C., Dettmering, D., Bosch, W., and Seitz, F.: DAHITI – an innovative approach for estimating water level time series over inland waters using multi-mission satellite altimetry, Hydrol. Earth Syst. Sci., 19, 4345-4364, 10.5194/hess-19-4345-2015, 2015.

Shamsudduha, M., Taylor, R. G., and Longuevergne, L.: Monitoring groundwater storage changes in the highly seasonal humid tropics: Validation of GRACE measurements in the Bengal Basin, Water Resources Research, 48, 10.1029/2011WR010993, 2012.

Van, T. D., Zhou, Y., Stigter, T. Y., Van, T. P., Hong, H. D., Uyen, T. D., and Tran, V. B.: Sustainable groundwater development in the coastal Tra Vinh province in Vietnam under saltwater intrusion and climate change, Hydrogeology Journal, 31, 731-749, 10.1007/s10040-023-02607-8, 2023.

---

## Author Comment (AC2)

**Responses to reviewer's comments**

We appreciate the constructive comments from Reviewer #2. In the revised manuscript, we have incorporated changes (highlighted in blue) to address all comments presented below in *italics*. Our responses to each comment are provided in **bold**.

1. Issues with scale: Have the authors thought about GRACEs coarse footprint? The basin, measuring about 23,000 km², is significantly less than GRACEs native resolution of around 90,000 to 300,000 km². Why is there the implication that interpolating GRACE to 0.25°−0.5° would add new detail? The ms explains that the "downscaled" delTWS on 0.25°−0.5° does not significantly change the results, since it correlates strongly with r≈0.85−1 with the initial 3° mascon and whole-Borneo signal. That is, the authors are basically looking at the same broad signal. Can we discuss whether this adds to our knowledge, or could this section be reworded or revised for simplicity? Could explain why these finer grids matter, as they are not independent signals.

R1: We thank the reviewer for these critical comments. We fully acknowledge that the study area is substantially smaller than the effective resolution of GRACE. We do not claim that the downscaled (0.25°-0.5°) GRACE data introduce new independent spatial information. The finer-grid analyses were used solely to: (1) examine internal consistency across commonly used Level-3 GRACE products at their distributed resolutions; (2) evaluate the degree of correlation between sub-mascon grids and larger-scale signals to test the limits of GRACE applicability in small coastal basins; and (3) quantify potential leakage effects from adjacent oceanic masses at different spatial scales. We have made revisions (lines 88-95, 268-272, 284-290 – see below) to clarify that the higher-resolution GRACE products do not provide independent ΔTWS signals but serve instead to assess spatial coherence and leakage behaviour across scales.

Lines 88-95: "This study examines whether GRACE can provide physically meaningful signals of groundwater storage variability in the small, coastal, and data-scarce Lower Kutai Basin (LKB), recognizing that its spatial scale lies below GRACE's effective resolution and is highly susceptible to ocean leakage. Specifically, we (1) compare  $\Delta$ TWS across multiple GRACE products, (2) evaluate the plausibility of GRACE-derived  $\Delta$ GWS against limited piezometric data, and (3) assess whether large-scale climate drivers, particularly ENSO, are detectable. The aim is to evaluate the limits of GRACE

in this challenging setting and to demonstrate that without a substantially expanded piezometric network, neither GRACE nor in situ observations alone can provide robust groundwater storage assessments to support the development of climate-resilient groundwater management strategies, particularly in rapidly urbanizing regions such as Nusantara where water security is a growing concern."

Lines 268-272: "Terrestrial water storage anomaly ( $\Delta$ TWS) values were classified into three spatial scales (Fig. 1b): (1) the study area grids, (2) a single 3° mascon grid, and (3) the entire Borneo Island. These classifications enable a diagnostic assessment of GRACE performance across spatial scales, recognizing that the finer (0.25°-0.5°) products are not independent of the native data. The analyses do not add spatial detail but rather test internal consistency among datasets and evaluate how leakage from adjacent ocean grids may influence  $\Delta$ TWS estimates in a small coastal basin that is below GRACE's effective footprint."

Lines 284-290: "Despite the differences,  $\Delta TWS$  in the study area remains highly correlated with both single-mascon and values for Borneo Island (r = 0.78-1; RMSE = 0.8-4.4 cm, Table S3). The study area tracks the single mascon closely (r = 0.93-1; RMSE = 0.8-2.5 cm) but divergence is more conspicuous relative to Borneo Island, particularly for COST-G (r = 0.78; RMSE = 1.3 cm). Applying leakage corrections increases coherence with the single mascon (r = 0.92 for GFZ, 0.94 for COST-G) yet reduces consistency with Borneo Island, especially for GFZ (r = 0.64; RMSE = 3.7 cm). The strong correlations among sub-mascon grids, the single mascon, and  $\Delta TWS$  values for Borneo Island suggest that the apparent fine-scale structure may not represent true spatial heterogeneity, underscoring the scale limitation of GRACE for basins smaller than its native resolution."

2. Data and uncertainty: The delGWS estimates are based on the subtraction of soil and surface water models from GRACE TWS and thus inherit all related uncertainties. You are using several GLDAS models and WGHM and reporting their differences. But it would be helpful to discuss the unreliability of those in this situation. For example, you noticed that one of the GLDAS runoff datasets was "implausible," and you discarded it – this says that model results can not be completely trusted. Did you compare GLDAS soil moisture or WGHM surface water with any local observations? If that is the scenario, it would be helpful if this could be mentioned. Your technique for producing 36 GWS "realizations" through combinations of GRACE and model ensembles is resourceful. Nevertheless, the selection to eliminate some 30% of them as "implausible". Have you thought about reducing model biases

instead? You are correct to have noted that these arithmetic anomalies mainly appear when delTWS is smaller than delSMS plus delSWS in the wet months, or the reverse during the dry months. Overlooking those months might affect your trends. It would be useful to have a numerical estimate of the effect of the filtering on the results. You show that removing the outliers improves the TWS–GWS correlation. It might be helpful to provide error estimates for delGWS or to note that the "mean plausible" series is just one model of noisy data? Assumptions of the model: Along these lines, it should be pointed out that GLDAS and WGHM dont account for groundwater pumping.

There appear to be signs that some wells are experiencing decline due to abstraction, but the GWS calculations dont account for this. If water pumping is stopped from the basin (and not simply recharging local surface water), GRACE would find a deficit. But the average GRACE DEL GWS trend indicates a very gentle positive change, even when one well went significantly deeper and had a local decrease. How do you resolve that? The conclusion blames the discrepancy on "withdrawals," but it would be nice to supply some quantification or at least investigate whether the pumping is too localized to be seen in GRACE. Are there other factors to take into account, such as groundwater draining from the basin by rivers or recharge assumptions?

R2: We appreciate the comments. We agree that uncertainties in both GRACE  $\Delta$ TWS and the simulated water storage components ( $\Delta$ SMS,  $\Delta$ SWS) may propagate to groundwater storage change ( $\Delta$ GWS) estimates. We have noted in the revised text that GLDAS and WGHM are not independently calibrated for our study area, as ground-based observations are largely unavailable (lines 601-605).

In the revised manuscript, we generated 54  $\Delta$ GWS realizations because we included two GRACE spherical harmonic products (GFZ and COST-G). We agree that discarding months with implausible  $\Delta$ GWS estimates can influence long-term trends. Therefore, we computed and found that the unfiltered ensemble mean  $\Delta$ GWS values produce a linear trend of 0.02 cm/year, whereas the filtered plausible values produce a larger trend of 0.2 cm/year (lines 425-427).

We agree that the ensemble mean plausible  $\Delta$ GWS estimates should not be interpreted as true values as validation using spatially distributed piezometric data and storage coefficients is necessary (lines 500-510). We also acknowledge that GLDAS and WGHM do not explicitly represent groundwater abstraction in the study area (lines 610-612).

Because spatially distributed storage coefficients that are required to convert groundwater level changes to  $\Delta GWS$  are unavailable, we can only estimate  $\Delta GWS$  using a range of possible storage coefficient values (lines 500-510). The effect of groundwater pumping on  $\Delta GWS$ , however, would only be apparent if it is spatially extensive and volumetrically large enough to influence basin-scale mass detected by GRACE. In our study area, abstraction is spatially concentrated south of Balikpapan City and at depths within specific screened intervals (lines 529-531). We consider that the observed declines in some piezometers are likely localized.

Lines 425-427: "The ensemble mean of plausible  $\Delta GWS$  ranges from -4.5 to 7.2 cm with an annual trend of 0.2 cm/year (p < 0.05), whereas incorporating implausible values results in  $\Delta GWS$  ranges between -5.6 and 8.7 cm with an annual trend of 0.02 cm/year (p > 0.05)."

Lines 500-510: "In the study area, groundwater abstraction is concentrated within ~20 km of coast, particularly south of Balikpapan City (Fig. S24). We compared ensemble mean plausible GRACE-derived  $\Delta$ GWS with groundwater level changes ( $\Delta$ GWL) as the storage coefficient (S or Sy) is not well constrained. Lithological logs reveal heterogeneous interbedded sand and clay units (Fig. S25; Arifin et al. (2024)). Piezometer depths range from 21 to 135 m; shallow screens up to ~40 m may represent unconfined aquifers, whereas deeper screens may tap confined aquifers. In similar deltaic settings such as the Mekong and Indo-Gangetic deltas, storage coefficients mostly vary from ~0.08 to 0.25 (mean ~0.15) for unconfined aquifers and from ~10-5 to  $8\times10^{-4}$  (mean ~5 $\times10^{-4}$ ) for confined aquifers (BGS and DPHE, 2001; Bonsor et al., 2017; Pechstein et al., 2018; Van et al., 2023). This uncertainty translates into a range of possible  $\Delta$ GWS values (Fig. 8). On average, confined aquifers yield ~0.5 mm of storage loss per metre of decline (S =  $5\times10^{-4}$ ), whereas unconfined aquifers yield ~15 cm (Sy = 0.15). Although correlations are unaffected by this uncertainty, the amplitude of GRACE-derived  $\Delta$ GWS remains highly uncertain without reliable storage coefficients."

Lines 529-531: "Groundwater abstraction is not regional but largely concentrated near the southern coast of Balikpapan City (Fig. S24), likely contributing to localized declines in groundwater levels."

Lines 601-605: "Since  $\Delta$ GWS is not directly measured by satellites but inferred through hydrological balance calculations, its accuracy depends on the reliability of  $\Delta$ TWS estimates and the simulated water components in the study area, primarily  $\Delta$ SMS and

 $\Delta$ SWS, which can propagate errors into  $\Delta$ GWS estimates. However, these simulated components are not locally calibrated as soil moisture, river stage, and lake volume observations are largely unavailable."

Lines 610-612: "Another major source of uncertainty in GRACE-derived ΔGWS calculations is the representation of surface water storage and anthropogenic influences in the study area, including groundwater abstraction within the study area in the GLDAS and WGHM datasets."

3. ENSO correlations: The moderate values of delTWS/delSMS versus ENSO indices seem to make sense and are in line with results from other studies, but care should be taken with the short time series. Did you test for statistical significance or account for autocorrelation? Has it been possible for you to investigate correlating detrended or deseasoned data? It might improve your case if you think about stripping the seasonal cycle before associating it with ENSO. In some way or other, the physical explanation works out: large El Niño events dry the basin, which in turn amplifies the GRACE delGWS drops. But it is really the soil moisture changes that are having an effect on TWS or it seems that delGWS is much less sensitive? It may be beneficial to emphasize how the soils and floodplains contribute significantly to the GRACE signal here.

R3: We thank the reviewer for these suggestions. In the revised manuscript, we have added p-values to indicate statistical significance (lines 453-467), all of which are statistically significant (p < 0.05). We have also computed correlations between climate indices and precipitation with deseasonalized and detrended  $\Delta$ TWS,  $\Delta$ SMS, and  $\Delta$ GWS. The resulting correlations are weaker than those obtained from the raw time series (lines 467-470), suggesting that ENSO-driven rainfall variability primarily influences seasonal changes in near-surface water storage components.

Lines 453-467: "Across GRACE products,  $\Delta$ TWS shows weak-to-moderate correlations (p-values <0.05) with ENSO indices, with r values ranging from -0.46 to -0.56 for MEI and -0.41 to -0.48 for ONI. The ensemble mean  $\Delta$ TWS yields correlations of -0.52 (MEI) and -0.46 (ONI), whereas weaker correlations are observed with the IOD (DMI, r = -0.26) and PDO (r = -0.25). These results indicate that ENSO is the dominant driver of  $\Delta$ TWS variability in the study area compared to IOD and PDO.

 $\Delta$ SMS exhibits slightly stronger ENSO sensitivity than  $\Delta$ TWS. Catchment and Noah LSMs yield r values with MEI and ONI ranging from -0.56 to -0.62, whereas the ensemble mean shows correlations of -0.60 (MEI) and -0.61 (ONI) with p-values <0.05. As with

 $\Delta TWS$ , correlations with DMI (-0.31 to -0.35) and PDO (-0.2 to -0.22) are weaker. These suggest that soil moisture anomalies are relatively sensitive to ENSO events. Monthly precipitation also correlates moderately with  $\Delta TWS$  (r = 0.47-0.52) and  $\Delta SMS$  (r = 0.46-0.58).

In contrast, correlations between ensemble mean plausible  $\Delta$ GWS in this study with climate indices or precipitation remain consistently weaker than those for  $\Delta$ TWS or  $\Delta$ SMS, with values of -0.41 (MEI), -0.33 (ONI), -0.18 (DMI), -0.23 (PDO), and 0.38 (precipitation) with p-values < 0.05. The ensemble mean plausible  $\Delta$ GWS estimates from GLWS datasets are similar, with values of -0.44 (MEI), -0.48 (ONI), -0.17 (DMI), -0.26 (PDO), and 0.29 (precipitation). In contrast, the GLDAS dataset exhibits substantially stronger correlations with MEI (-0.6) and ONI (-0.63), whereas correlations with other indices are relatively comparable: -0.32 (DMI), -0.21 (PDO), and 0.43 (precipitation)."

Lines 467-470: "In addition, we repeated the correlation analysis using deseasonalized and detrended components of  $\Delta TWS$ ,  $\Delta SMS$ , and  $\Delta GWS$  which resulted in generally weaker correlations than those of the raw data (Fig. S23). This reduction indicates that much of the ENSO-related signal is expressed through modulation of the seasonal cycle of water storage changes rather than non-seasonal anomalies."

4. Piezometer comparisons: I do like the attempt to incorporate groundwater wells, but please be careful. In the first place, matching GRACE delGWS (in cm over the whole basin) with a small set of point measurements is of course an approximation. Your conclusion shows a "moderate" correlation, which is reasonable. Claiming, however, that the time series "align with groundwater-level" (Abstract) goes too far. For example, GRACE indicates a small upward trend in general, while the deep well has a downward trend. How are the authors confirming that these are actually deep observation wells? Furthermore, head to storage conversion demands a certain yield values, which has not been discussed. Thus, the comparison is qualitatively of the rise/fall type instead of being concerned with quantitative volume. Please the range of specific yield you are assuming, or flagging this as a possible source of uncertainty? without knowing aquifer properties, its not easy to relate a GRACE-derived cm change to an observed m depth change.

R4: We thank the reviewer for these comments. We agree that matching basin-scale GRACE-derived  $\Delta$ GWS with a limited number of point-scale groundwater level observations is inherently approximate. We have revised the abstract accordingly. As in R2 (lines 500-510), we have reported the depths of screened intervals in each piezometer to indicate that observations are generally from deep. We agree that

converting head changes to storage change requires spatially distributed storage coefficients, which are not available in this basin. We have estimated  $\Delta GWS$  from piezometric data using plausible storage coefficients derived from comparable deltaic aquifers (lines 500-510).

Abstract: "Groundwater is considered a climate-resilient source of freshwater yet its long-term response to climate variability remains poorly understood in environments with limited ground-based monitoring networks. In the Lower Kutai Basin where Indonesia's new capital (Nusantara) is under development, we examine evidence from Gravity Recovery and Climate Experiment (GRACE) satellite data, global-scale models, precipitation records, and in situ piezometric observations to investigate groundwater storage changes (ΔGWS) over the last two decades. GRACE-derived terrestrial water storage anomalies (ΔTWS) exhibit strong seasonal and interannual variability that are consistent across different spatial scales (r = 0.78-1) and are dominated by changes in soil moisture storage (ASMS). Paired land-ocean grid analyses show productdependent residual correlations after detrending and deseasonalizing with values of up to 0.68, suggesting potential ocean leakage. Across 54 realizations, 21-60% of ΔGWS estimates per realization are plausible with ensemble mean values ranging from -4.5 to 7.2 cm. Agreement between GRACE-derived AGWS and groundwater-level anomalies (AGWL) varies by site and depth, with correlations that are generally weak and reflect discrepancies in scale between GRACE's basin-scale signals and localized aguifer dynamics influenced by heterogeneity and groundwater abstraction. Statistical analyses show weak-to-moderate coupling of  $\Delta TWS$  and  $\Delta SMS$  with ENSO indices (r = -0.4 to -0.6), whereas ΔGWS is less responsive. The strongest 2015-2016 El Niño is a notable example, associated with  $\Delta TWS$  deficits (-2.4 to -4.6 cm/month) and  $\Delta GWS$ declines (-1.1 cm/month). High-frequency (hourly) groundwater-level observations indicate that episodic, high-intensity rainfall events (>90th disproportionately contribute to groundwater recharge. These findings underscore the need for expanded in situ monitoring and accurate storage coefficients to validate GRACE-derived AGWS, particularly in regions such as Nusantara where water security is a growing concern."

5. Monitoring groundwater: Filtering for the minimum daily level is an able strategy and seems to track recharge. As a reader, I am interested in the strength of that: can pump-off intervals be incorrectly interpreted, or can the lowest daily level nevertheless be influenced by slower pumping rates? Can you indicate how well these loggers are calibrated? A mention of data quality control, would be helpful.

R5: We appreciate this comment and agree that the filtering approach may involve some uncertainties. In the revised manuscript, we have added a brief description of the procedure used to ensure data quality (lines 707-716).

Lines 707-716: "The recorded data were visually inspected for noise, gaps, and abrupt shifts that may indicate sensor malfunction. The pressure transducers were factory-calibrated and manual depth-to-water checks were carried out at deployment and retrieval for calibration. Barometric pressure corrections were applied to all measurements. By filtering out data recorded during pumping periods and retaining only the minimum daily groundwater levels from recovery phases, an approximated near-static groundwater level time series was extracted from the pumping well (Fig. 9b). These suggest that with appropriate data filtering techniques, pumping wells can serve as a viable and cost-effective alternative to traditional monitoring wells, offering a scalable solution for groundwater monitoring in Indonesia. However, reliability requires periodic manual validation of piezometric sensors, such as monthly checks during the first few months of deployment and semi-annual checks thereafter, in order to mitigate instrumental issues such as sensor drift and calibration errors that can bias long-term records at seasonal or interannual timescales."

6. Clarity and flow: The writing is generally easy to follow, although there are spots where its a bit dense. The methods section is very thorough, although it could do with more obvious labelling. It may be a good idea to divide the GRACE discussion from the GLDAS/WGHM. Some of the sentences are also very long – try breaking them up to make them easier to read, especially in the introduction and methods sections.

R6: We appreciate the positive feedback. We have reviewed and revised the prose of the manuscript to improve its clarity and flow.

7. This paper shows a worthy effort in tackling a difficult problem, however, I am concerned about the 23,000 km² study area. The authors have used a wide range of data and monitoring methods. I would recommend significant revision, moderating any very assertive conclusions, especially about delGWS match with wells etc, and being forthright about the large uncertainties involved. I look forward to reading a revised version.

R7: We thank the reviewer for the constructive comments. We acknowledge that the study area is small relative to GRACE's native footprint. Accordingly, we have reviewed

all comments and made revisions to present a balanced interpretation consistent with GRACE's scale and uncertainty limitations.

**References**

- Arifin, Shamsudduha, M., Ramdhan, A. M., Reksalegora, S. W., & Taylor, R. G. (2024). Characterizing deep groundwater using evidence from oil and gas exploration wells in the Lower Kutai Basin of Indonesia. *Hydrogeology Journal*, 32, 1125-1144. https://doi.org/10.1007/s10040-024-02776-0
- BGS, & DPHE. (2001). Arsenic contamination of groundwater in Bangladesh. In D. G. Kinniburgh & P. L. Smedley (Eds.), *British Geological Survey Technical Report WC/00/19*. British Geological Survey.
- Bonsor, H. C., MacDonald, A. M., Ahmed, K. M., Burgess, W. G., Basharat, M., Calow, R. C., Dixit, A., Foster, S. S. D., Gopal, K., Lapworth, D. J., Moench, M., Mukherjee, A., Rao, M. S., Shamsudduha, M., Smith, L., Taylor, R. G., Tucker, J., van Steenbergen, F., Yadav, S. K., & Zahid, A. (2017). Hydrogeological typologies of the Indo-Gangetic basin alluvial aquifer, South Asia. *Hydrogeology Journal*, *25*(5), 1377-1406. https://doi.org/10.1007/s10040-017-1550-z
- Pechstein, A., Hanh, H. T., Orilski, J., Nam, L. H., & Manh, L. V. (2018). Detailed Investigations on the hydrogeological situation in Ca Mau Province, Mekong Delta, Vietnam. <a href="https://www.recyclingrohstoffe-dialog.de/EN/Themen/Wasser/Projekte/abgeschlossen/TZ/Vietnam/techn\_repIII-5\_en.pdf?\_blob=publicationFile&v=3">https://www.recyclingrohstoffe-dialog.de/EN/Themen/Wasser/Projekte/abgeschlossen/TZ/Vietnam/techn\_repIII-5\_en.pdf?\_blob=publicationFile&v=3</a>
- Van, T. D., Zhou, Y., Stigter, T. Y., Van, T. P., Hong, H. D., Uyen, T. D., & Tran, V. B. (2023). Sustainable groundwater development in the coastal Tra Vinh province in Vietnam under saltwater intrusion and climate change. *Hydrogeology Journal*, *31*(3), 731-749. https://doi.org/10.1007/s10040-023-02607-8